

# Use of GPCC and GPCP Precipitation Products and GRACE and GRACE-FO Terrestrial Water Storage Observations for the Assessment of Drought Recovery Times

Çağatay Çakan[1], M. Tuğrul Yımaz[1], Henryk Dobslaw[2], E. Sinem Ince[3], Fatih Evrendilek[4], Christoph Förste[3], Ali Levent Yagci[5]

[1]Department of Civil Engineering, Water Resources Laboratory, Middle East Technical University, Ankara 06690, Turkey
[2]Section 1.3: Earth System Modeling, GFZ German Research Centre for Geosciences, Potsdam 14473, Germany
[3]Section 1.2: Global Geomonitoring and Gravity Field, GFZ German Research Centre for Geosciences, Potsdam 14473, Germany
[4]Department of Civil and Environmental Engineering, University of Maine, Orono, Maine 04469, USA
[5]Department of Geomatics Engineering, Gebze Technical University, Kocaeli 41400, Turkey

*Correspondence to*: Çağatay Çakan (cakan.cagatay@metu.edu.tr)

**Abstract.** Meteorological and hydrological processes depend on accurate precipitation observations. Most precipitation products utilize station-based observations directly or to bias correct satellite retrievals. Thus, the validation of station-based precipitation products requires further independent data. This study aims to assess the accuracy of the Global Precipitation Climatology Center (GPCC) and Global Precipitation Climatology Project (GPCP) precipitation products by estimating hydrological drought recovery time (DRT) from terrestrial water storage anomaly (TWSA) acquired from satellite gravimetry and the required precipitation amount across the five main Köppen-Geiger climate zones. Station-based precipitation products, namely GPCC Full Data Monthly Product v2022 and GPCP v3.2 Monthly Analysis Product, were utilized to estimate DRT. Additionally, the JPL mascon and G3P Total Water Storage (TWS) monthly-solutions from the Gravity Recovery and Climate Experiment (GRACE) and GRACE Follow-On (GRACE-FO) satellite missions were also employed for the DRT estimation. DRT was estimated through the following two methods: (1) storage deficit, determined as the negative residual of detrended TWSA from its climatology, and (2) required precipitation amount, derived from the linear relationship between cumulative detrended smoothed precipitation anomaly (cdPA) and detrended TWSA. The results show no significant differences in the mean DRT estimations using GPCC and GPCP. Conversely, DRT estimation using JPL mascon is 2.6 months longer on average than that using G3P. The equatorial zone showed the shortest DRT estimation, 10.3 months, while the polar zone had the longest, 16.2 months. Except for the polar zone, the arid zone shows the highest DRT estimations, 13.9 months. Consistency in DRT estimations between the two methods was high across the different climate zones, with the equatorial zone exhibiting the highest, 97.8%, and the polar zone the lowest, 74.9%. Similar to mean DRT estimation results, the differences in consistency were not significant for the estimations obtained from GPCC and GPCP. In contrast, the G3P showed approximately 5.0% higher consistency than the JPL mascon. The findings based on DRT estimations indicate a close agreement between GPCC and GPCP. Moreover, G3P was more consistent in DRT estimation with precipitation products than





JPL mascon. These results provide necessary information for precipitation and TWSA product accuracy by using the hydrological drought characteristics, which helps in understanding the meteorological and hydrological processes.

## 1 Introduction

Precipitation is a pivotal element in the global water cycle. It provides freshwater to continental regions and thereby allows vegetation to flourish. Average precipitation amounts and the associated temporal distribution of rain events characterize climate zones and terrestrial ecosystems (Bayar et al., 2023; Lai et al., 2018). Too much or too little precipitation than usual, however, can have very severe impacts on biosphere, agriculture, and human societies in general. The close monitoring of droughts (Barker et al., 2016; Lai et al., 2019; Wu et al., 2023; Xu et al., 2015) and floods (Belabid et al., 2019; Harris et al., 2007; Maggioni & Massari, 2018), as well as the prediction of precipitation at short, medium and long forecast horizons (Akbari Asanjan et al., 2018; Senocak et al., 2023) are a central objective of hydrometeorologic research.

In situ observations from rain gauges are typically utilized to monitor precipitation (Barker et al., 2016; Wehbe et al., 2017; Wei et al., 2019). However, the distribution of gauge stations is often sparse and uneven, particularly over complex terrains where stations may be difficult to install and maintain (Wang et al., 2017). In contrast, satellite and satellite blended based precipitation products derived from remote sensing instruments have made essential strides, offering varying spatiotemporal resolutions as a viable alternative to ground-based observations (Bai et al., 2019; Prakash et al., 2015; Wang et al., 2017; Wu et al., 2023). The Global Precipitation Climatology Center (GPCC) and Global Precipitation Climatology Project (GPCP) are frequently used precipitation products with global coverage (Adler et al., 2003; Sun et al., 2018). GPCC represents ground-based precipitation observations, whereas GPCP is a combination of satellite and in situ station observations.

Products from both GPCC and GPCP have been frequently compared with each other and against a wide range of atmospheric reanalyses (e.g., Prakash et al., 2015). At regional scales, particularly in the tropics, good agreement has been observed between GPCC and GPCP (Negrón Juárez et al., 2009; Sun et al., 2018). Moreover, the spatial distribution of annual and seasonal rainfall climatology across West Africa was found to be consistent between GPCC and GPCP (Lamptey, 2008). Although there are regional similarities, there are also distinct differences. GPCC outperformed GPCP against station-based precipitation data in China (Wang et al., 2017), demonstrated enhanced spatiotemporal representativeness of precipitation patterns in Iran (Darand & Khandu, 2020), and showed superior performance in the Sahel region based on statistical error metrics (Ali et al., 2005). In general, these studies evaluate precipitation products by comparing them with in-situ observations. However, given both datasets utilize ground station-based observations, indecency of the evaluation analyses becomes an important aspect. Accordingly, additional independent assessments may be needed for precipitation products that utilize observations such as GPCC and GPCP.





Drought monitoring is crucial since drought is one of the most destructive disasters, resulting from a significant decrease in a
region's water resources over an extended period. It can have disastrous consequences for ecosystems, human health,
agriculture, irrigation, and water supply (AghaKouchak et al., 2015; Ding et al., 2020; Mishra & Singh, 2010; Patz et al., 2014;
Piao et al., 2010). Drought indices, such as the standardized precipitation index (SPI, Mckee et al., 1993) and the standardized
precipitation evapotranspiration index (SPEI, Vicente-Serrano et al., 2010), the standardized runoff index (SRI, Shukla &
Wood, 2008) and the standardized streamflow index (SSI, Vicente-Serrano et al., 2012) are utilized to characterize drought
characteristics (e.g., frequency, severity, and recovery time). Meteorological droughts arise from insufficient precipitation,
whereas hydrological droughts result from insufficient water storage (Behrangi et al., 2015; Keyantash & Dracup, 2002;
Thomas et al., 2014). SPI focuses solely on precipitation data, while SPEI utilizes precipitation and evapotranspiration data.
SSI hinges on runoff yield from the land surface, while SRI utilizes streamflow in river channels (Lai et al., 2019). Complex
hydrological models utilize precipitation data for hydrological drought assessment based on SSI and SRI (Lai et al., 2018;
Madadgar & Moradkhani, 2014). Alternatively, the water storage deficit can provide insights into hydrological drought without
employing elaborate hydrological models (Thomas et al., 2014). It only requires measurements of the amount of water stored
at or underneath the ground and is employed in the estimation of drought recovery time (DRT). By combining precipitation
and terrestrial water storage (TWS) observations it is even possible to predict the amount of precipitation that will be required
to re-fill any storage deficit (Singh et al., 2021).

The satellite mission Gravity Recovery and Climate Experiment (GRACE) provides such measurements of TWS (Springer et
al., 2017). GRACE was conducted jointly by the National Aeronautics and Space Administration (NASA) and the German
Aerospace Center (DLR) from 2002 until 2017. Since 2018, GRACE Follow-On (GRACE-FO), successor of GRACE, is
operated by NASA together with the German Research Centre for Geosciences (GFZ) to further extend the data record until
present. Terrestrial water storage anomalies (TWSA), encompassing all subsurface and surface water balance components, are
obtained by measuring tiny irregularities in the orbits of two identical twin-satellites that are trailing each other with a distance
of roughly 200 km in polar orbit of initially 490 km altitude (Wahr et al., 2004). Temporal changes in the Earth's gravity field
are computed from the comparison of observations from different times. Once atmospheric, oceanic and geophysical effects
are removed, the remaining signal on monthly-to-interannual scales reflects variations in TWS. The ready-to-use TWS data
from GRACE and GRACE-FO missions are made available either as spherical harmonic (SH) or mass concentration solutions
(mascons). GRACE-based TWS has been used in the past to relate interannual variations in TWS to large-scale climate modes
(Pfeffer et al., 2023), and to validate hydrological models (Döll et al., 2024). There were even attempts to assimilate GRACE
data into land-surface schemes (Eicker et al., 2014; Tangdamrongsub et al., 2021). Hence, GRACE and GRACE-FO are
currently the most used datasets in global TWS.

The use of GRACE and GRACE-FO TWS products could be used for an independent assessment of precipitation products by
drought monitoring, serving as an alternative to assessments conducted with hydrological models (Beck et al., 2017;





Gebrechorkos et al., 2024). Existing studies evaluate precipitation products using drought monitoring and focus on drought indices such as SPI and SPEI (Golian et al., 2019; Wei et al., 2019, 2021). However, more independent assessment studies using key parameters that encompass all subsurface and surface water balance components, such as TWS, are necessary to better understand the utility of precipitation products. This is particularly important for hydrological drought assessment, as spatial variability across different climate zones globally is still inadequately explored.

The objective of this study is to independently evaluate and compare the GPCC and GPCP precipitation products by using the GRACE and GRACE-FO TWS data (i.e., the JPL mascon and G3P products) in terms of assessing drought conditions. By estimating DRT based on TWSA and required precipitation amount, the present study compares the suitability of these precipitation products for global hydrological applications across different climate zones as defined by the Köppen-Geiger classification. This comparative analysis provides a basis for understanding the relationship between hydrological droughts and global precipitation products through DRT estimations.

## 2 Methodology

### 2.1 Datasets

### 2.1.1 GPCC and GPCP Precipitation

Established in 1989 by the World Meteorological Organization (WMO), the Global Precipitation Climatology Center (GPCC) integrates monthly land precipitation data from various sources including the global telecommunication systems (GTS), synoptic weather reports (SYNOP), and monthly climate reports (CLIMAT). GPCC offers different precipitation products with varying spatiotemporal resolutions, such as the Full Data Monthly Product (GPCC FDM), the Monitoring Product, and the First Guess Monthly Product. Given its suitability for model verification and water cycle studies, this study utilized GPCC FDM v2022 (Schneider et al., 2022) to analyze the relationship between precipitation and TWS (Schneider et al., 2014). The GPCC FDM dataset provides monthly precipitation data at a spatial resolution of 0.5° from 1891 to 2020. The GPCC data were sourced from the Deutscher Wetterdienst (German Meteorological Service) website (https://opendata.dwd.de/climate_environment/GPCC/html/fulldata-monthly_v2022_doi_download.html).

The Global Precipitation Climatology Project (GPCP) is a combined satellite-gauge precipitation product overseen by the World Climate Research Program (WCRP) under its Global Water and Energy Experiment (GEWEX) Data and Assessment Panel (GDAP). It integrates rain gauge observations with satellite data to generate global precipitation estimates. For this study, we utilized GPCP v3.2 Satellite-Gauge (SG) Combined Data (Huffman et al., 2023). The monthly GPCP v3.2 dataset spans from 1979 to present at a spatial resolution of 0.5°. The data are available from the Goddard Earth Sciences Data and Information Services Center (https://disc.gsfc.nasa.gov/datasets/GPCPMON_3.2/summary).





### 2.1.2 TWS from GRACE/GRACE-FO

We analyzed the GRACE/GRACE-FO Level-3 products of the G3P (Güntner et al., 2023) and the JPL Release 6 mascon (Watkins et al., 2015; Wiese et al., 2023) TWS datasets to estimate water storage deficit. TWS comprises the sum of snow, ice, surface water, soil moisture, and groundwater. The G3P TWS data were acquired from the GFZ Information System and Data Center (ftp://isdcftp.gfz-potsdam.de). The JPL mascon TWS data were downloaded from the Virtual Directories of Earth Data CMR (https://cmr.earthdata.nasa.gov/virtual-directory/collections/C2536962485-POCLOUD/temporal/2002/04/16).

Both G3P and JPL mascon offer a higher spatial resolution (0.5°) than do the spherical harmonic solutions and monthly data. However, datasets suffer from missing monthly data, particularly after 2011, due to satellite battery issues. To ensure consistent comparisons between continuous TWS and precipitation time series data, we filled the missing months in the time series by averaging the data from the previous and subsequent two months, resulting in a mean of four months (Andrew et al., 2017; Long et al., 2015). However, there is also a time gap between the GRACE and GRACE-FO missions, spanning from July 2017

(i.e., the end of the science phase of the GRACE mission) to May 2018 (i.e., the launch of GRACE-FO). This period is left missing.

The JPL mascon TWS dataset represents anomalies relative to a long-term mean from January 2004 to December 2009, while the G3P TWS dataset uses a long-term mean from April 2002 to December 2020 as the baseline. To ensure consistent

comparisons between the time series, we adjusted the baseline of the JPL mascon TWS to match that of the G3P TWS (Humphrey et al., 2023; Monthly Mass Grids - Global Mascons (JPL RL06.1_v03) | Data Portal – GRACE Tellus, n.d.). This involved averaging each grid point over the period of April 2002 to December 2020 and subtracting it from the entire time series. Thus, this study utilized monthly TWS data from April 2002 to December 2020 for DRT analysis.

### 2.1.3 Köppen-Geiger Climate Classification

Globally, the Köppen-Geiger climate classification system, based on temperature and precipitation, is widely used for regional climate zonation by a diverse range of disciplines, such as climate research, physical geography, hydrology, agriculture, biology, and education (Kottek et al., 2006). In response to the need for current and well-documented global climate classifications, a new Köppen-Geiger climate map was released for a high-resolution (0.5°) depiction of global climates for the 1951-2000 period (Kottek et al., 2006). The present study utilized a more recent version of this dataset covering the 1986-

2010 period at a higher spatial resolution of 0.083° (Rubel et al., 2017). To achieve consistency with our TWS and precipitation data (grid resolution of 0.5°), the Köppen-Geiger climate classifications were re-gridded using bilinear interpolation. This study focused on the following five main Köppen-Geiger climate categories: equatorial, arid, warm temperate, snow, and polar, as illustrated in Fig. 1. The Köppen-Geiger climate classification scheme used in this study is available at https://koeppen-geiger.vu-wien.ac.at/present.htm.





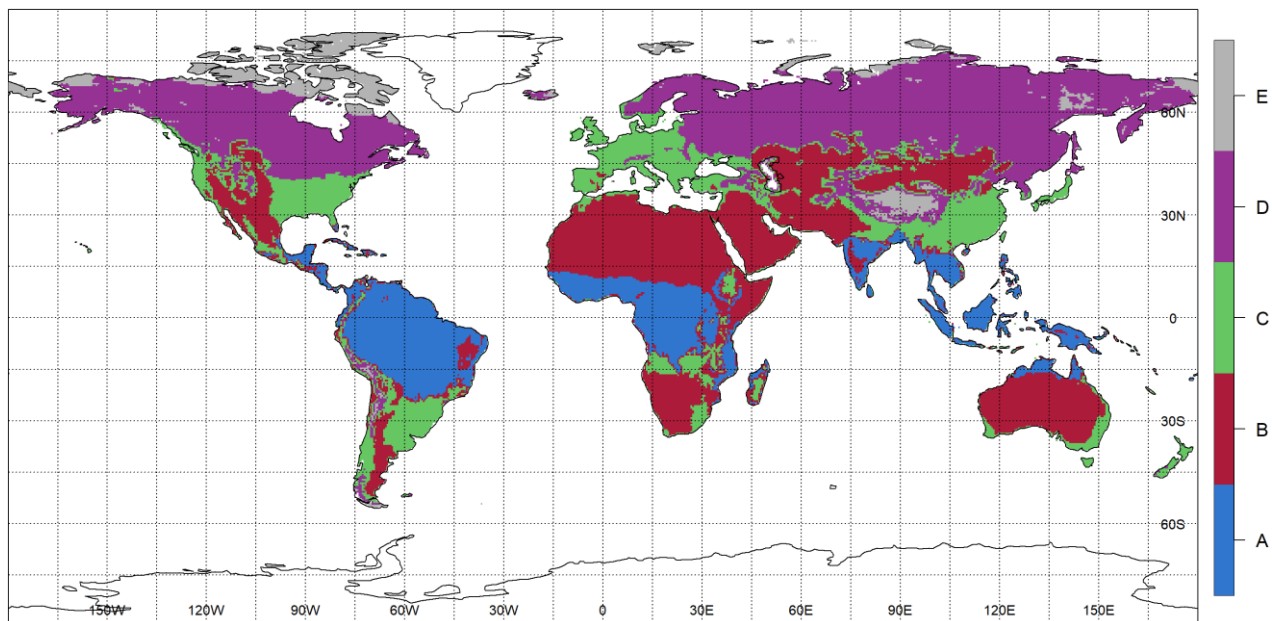


**Figure 1. Köppen-Geiger climate zone classification with the five main zones Equatorial (A); Arid (B); Warm Temperatures (C); Snow (D); and Polar (E).**

## 2.2 Water Balance Equation

TWS changes are closely related to precipitation via the water balance equation.

$ds/dt = P - ET - R$ ,                        (1)

where ds/dt is storage change with time, P is precipitation, ET is evapotranspiration, and R is runoff, all usually given in mm equivalent water height per month (mm/month).

Any storage change with time (*ds/dt*) must be caused by a water flux, which might be vertically between surface and atmosphere (P or ET). Horizontal fluxes at or underneath the Earth's surface is summarized as lateral runoff (R). Since gravity

missions directly observe TWSA relative to a (undefined) long-term mean value, it can be used to derive information about water fluxes at a wide range of timescales. In principle, the difference between two storage estimates separated by 30 days (the usual sampling of the GRACE solutions) allows to derive quantitative information about the amount of precipitation that occurred during this time span.

## 2.3 Deviation of Storage (dTWSA)

Understanding the magnitude of water deficits remains crucial for determining drought recovery timelines. These water deficits can be directly inferred from the variability in TWSA data (Thomas et al., 2014). Variations in water storage can be influenced by long-term factors, such as glacier mass accumulation and/or groundwater extraction. To isolate the impact of such long-



term processes, we detrended the TWSA data for each grid. Detrending removes the linear trend, essentially isolating the deviations from this long-term trend, which we referred to as deviations of storage (dTWSA).

### 180  2.4 Cumulative Detrended Precipitation Anomaly (cdPA)

To ensure consistency with dTWSA, we perform a temporal integration of the precipitation data. Then, cumulative precipitation anomalies (cPA) were obtained by subtracting the mean precipitation observed from April 2002 to December 2020 (reference period) from the actual precipitation data for each product:

$$cPA_{x,y} = cP_{x,y} - \overline{cP_{x,y}}, \tag{2}$$

where $cP_{x,y}$ is cumulated precipitation at $x,y$ grid point, and $\overline{cP_{x,y}}$ is the temporal mean of the cumulative precipitation at $x,y$ grid point.

Similar to dTWSA, we smoothed the cumulative precipitation anomalies (scPA) derived from GPCC and GPCP by applying a 3-month moving average filter (Singh et al., 2021). This process effectively reduced short term fluctuations or noise in the

anomaly data. To isolate variations in precipitation anomalies from long-term trends, we further detrended the smoothed precipitation anomalies. This additional step reduced any remaining long-term trends in the precipitation patterns. Finally, the cumulative detrended precipitation anomaly (cdPA) data were obtained.

$$cdPA_{x,y} = scPA_{x,y} - trend(scPA_{x,y}), \tag{3}$$

where $scPA_{x,y}$ is the smoothed precipitation anomalies at $x,y$ grid point, and $trend(scPA_{x,y})$, is the trend of the smoothed

precipitation anomalies at $x,y$ grid point.

### 2.5 Relationship between dTWSA and cdPA

Variations in the key water fluxes (ET, R, and P) cause fluctuations in TWS, a crucial component of the water budget equation. This study leveraged the assumption of a constant relationship between precipitation and the combined evapotranspiration and runoff (ET+R) flux. This assumption allows us to infer potential variations in precipitation based on changes in TWSA.


This predictive model enables us to estimate the amount of precipitation necessary to balance a water storage deficit. This approach offers a valuable tool for understanding and managing water resources by directly linking precipitation dynamics to changes in TWSA. A linear relationship between dTWSA and cdPA was established to estimate the required precipitation amount as follows:

$$cdPA = \beta_0 + \beta_1 * dTWSA + \varepsilon, \tag{4}$$

where $\beta_0$ is the intercept; $\beta_1$ is the slope (regression coefficient); and $\varepsilon$ represents the residual errors of the fit.



The units of cdPA and dTWSA are both in mm/month. Therefore, a $\beta_1$ value 1 signifies that cdPA is well represented by dTWSA. In regions where $\beta_1$ equals 1, precipitation changes directly translate to storage variations. Conversely, regions with

$\beta_1$ greater than 1 indicate that some of the local precipitation is immediately transported away and does not change the local storage. This suggests that the variability in storage data only partially captures all precipitation due to other hydrological processes (e.g., ET and R) in these regions. Regions with $\beta_1$ less than 1 suggest that a smaller amount of precipitation than needed is sufficient to address the storage deficit. In other words, there must be either additional inflow from other places that is phase-locked with local rain events or severe positive biases in rainfall as seen by GPCC or GPCP, leading to an

underestimation of the precipitation required based solely on storage variations (Singh et al., 2021).

Following the study of Singh et al. (2021), in addition to regression analyses (i.e., $\beta_0$ and $\beta_1$), we have also calculated the correlation coefficient ($\rho$, between cdPA and dTWSA) and maximum drought length over each pixel utilizing 19 years of monthly data (i.e., between 2002 and 2020). Here, a positive relationship is expected between cdPA and dTWSA such that

positive (negative) precipitation anomalies should lead to increased (decreased) storage changes. Reverse signs in cdPA and dTWSA anomalies indicate weak or no linear relationship between the two variables. Accordingly, the study of Singh et al. (2021) eliminated the pixels that contain weak or no linear relationship between cdPA and dTWSA (i.e., $\rho < 0$, $\beta_1 < 1$, and maximum drought length $< 5$ months) from the global analyses. Sampling errors could cause fluctuations around 1, and random variability may cause some of the pixels to have $\beta_1$ values slightly less than 1, while these pixels may still have some

considerable linear relationship. Accordingly, while the methodology of Singh et al. (2021) masked out regions with $\beta_1 < 1$, in this study, regions with $\beta_1 < 0$ are masked out in addition to regions with $\rho < 0$ and maximum drought length $< 5$ months.

## 2.6 DRT

Both TWSA and P will be used to quantify DRT. We closely follow the methodology given by Singh et al. (2021) and utilize two different methods to estimate DRT. The first method, based on storage deficit, utilizes only GRACE data to quantify DRT

as the duration of the residuals of TWSA from its climatology. The second method is based on the required precipitation amount derived from both TWSA and precipitation datasets. In this approach, a drought is considered to end when the absolute required precipitation amount surpasses the observed precipitation amount. We will thus utilize two different precipitation products (GPCC and GPCP), two different GRACE storage estimates (G3P and JPL mascon), and two different DRT estimation methods (storage deficit and required precipitation amount) to study the consequences of those processing choices

for different climate zones as defined by the Köppen-Geiger climatology (In total, 8 different DRT will be calculated.).

### 2.6.1 DRT based on Storage Deficit

The deviation of TWSA from its climatology can offer valuable insights into drought characteristics. To calculate this deviation, we first create a reference point by averaging TWSA values for each month across the entire time series. For example, to establish the average January TWSA, we would calculate the mean of all January values in the data. This average





monthly TWSA represented the climatology for that specific month. Subsequently, we calculated how much each TWSA data point deviated from this average climatology by subtracting the corresponding monthly climatology value. To identify drought events, we first calculated a climatology (long-term average) for TWSA data from both JPL mascon and G3P products using the time series from April 2002 to December 2020. This climatology serves as a reference point for typical TWSA conditions for each month. We then subtracted the corresponding monthly climatology value from each TWSA data point, resulting in

residuals. Negative residuals indicate water storage deficits (Thomas et al., 2014). We classified periods with persistent negative residuals lasting longer than three consecutive months as drought events, signifying prolonged periods of below-average water storage (Singh et al., 2021). Negative residuals lasting less than three consecutive months were not classified as droughts. However, if a new negative residual period began within one month of a previous drought recovery, we considered them as a continuation of the same drought event. This approach ensured a cohesive record of drought occurrences over time.

By applying these criteria, we were able to establish a comprehensive inventory of drought characteristics for each grid point. This inventory served as the basis for our DRT estimation using the storage deficit method. This method analyzed the duration of negative residuals (i.e., smaller storage values than usual) of dTWSA at each location and time, thereby providing insights into the temporal patterns and the severity of drought events.

### 2.6.2 DRT based on Required Precipitation Amount

The required precipitation amount is obtained from the linear relationship between dTWSA and cdPA. The storage deficit amount is represented by dTWSA, while cdPA, the output of this relationship, represents the required precipitation amount. To quantify the absolute required precipitation amount, the climatology of precipitation over each pixel was added back into the estimated required precipitation amount. The DRT estimation was then conducted by analyzing the duration during which the observed precipitation amount exceeded the absolute required precipitation amount for any given time and location (Singh

et al., 2021). This approach allows for a comprehensive assessment of DRT dynamics across the different regions and periods.

### 2.7 Accuracy Analysis

### 2.7.1 Consistency in DRT Estimations

The consistency (level of agreement) between the two DRT estimations was quantified by assessing the differences in the timing obtained from both methods. In this context, consistency referred to the temporal difference between the estimated

DRTs from each method, as categorized in Table 1. For example, if the time difference between the two methods fell within 1 or 2 months, the location was categorized as consistency category 1. By comparing the time differences between the DRT estimates from each method for each TWS-precipitation product, we were able to quantify the consistency between the two approaches. This analysis provides valuable insights into the reliability and robustness of the DRT estimations. In essence, it helped us assess how well the two methods converged on similar DRT values for the same locations.




**Table 1. Consistency categories in DRT estimations**

| Consistency category | Time difference (months) |
|:---:|:---:|
| 1 | 1-2 |
| 2 | 3-4 |
| 3 | 5-8 |
| 4 | 9+ |

### 2.7.2 Calculated Statistics

In our analyses, we use standard error (SE) as a measure of the uncertainty associated with the means of our datasets. A smaller SE value indicates a more precise estimate of the mean DRT over each pixel, which is often achieved with less variation in the data (Lee et al., 2015). The value of SE was separately calculated for each grid point and climate zone as follows:

$$SE_{x,y} = \frac{SD_{x,y}}{\sqrt{n_{x,y}}}, \tag{5}$$

where $SD_{x,y}$ is the standard deviation of DRT at $x,y$ grid point, and $n_{x,y}$ is the length of the dataset at $x,y$ grid point.

In addition to SE, we employed confidence intervals (CIs) to assess the uncertainty around the mean values of our datasets (Altman & Bland, 2005; Curran-Everett, 2008; Lee et al., 2015). CIs provide a range of values that are likely to contain the true population mean with a specified level of confidence (95% in our case) as follows:

$$CI_{x,y} = \mu_{x,y} \mp 1.96 * SE_{x,y}, \tag{6}$$

where $\mu_{x,y}$ is the mean DRT at $x,y$ grid point, and SE is the standard error of DRT at $x,y$ grid point.

## 3 Results

### 3.1 Relationship between cdPA and dTWSA

Figure 2a illustrates the spatial distribution of correlation coefficients between dTWSA and cdPA for one selected data combination of dTWSA from G3P and cdPA from GPCP (from now on abbreviated as G3P&GPCP). G3P&GPCP was selected to show actual values since the coupled product has the highest correlation coefficient (0.30). We find high correlations over Australia (0.55), South America (0.46) and south Africa ($\rho > 0.47$), where both water storage variations are substantial and in situ observing networks are dense. The negative correlations over polar regions (~70% of grids), where water storage decline is strong during and after the melting season without any direct relation to the incoming precipitation, are more than the other regions (~10% of grids). Similar disagreements are found in highly arid climates in Northern Africa and Central Asia, where water storage variations are tiny and GRACE observations are very likely dominated by measurement noise.

In addition, Figures 2b-d show the spatial distribution of differences in these correlation coefficients for the other possible combinations relative to the results obtained for G3P&GPCP. When switching to JPL mascon (Figures 2c), we see differences





primarily in arid climates with generally smaller TWS variations and a consequently poor signal-to-noise ratio in the GRACE data, where processing choices (like spatially variable a priori constraints as applied in the mascon) do have a greater effect. Switching from GPCP to GPCC (Figs. 2c and 2d) affects correlations to a much larger extent in many more regions, in particular over places with less dense in situ networks since the standard deviation of the difference in Figs. 2d (0.21) is higher

than those of difference in Figs. 2c (0.14). Higher correlations for GPCP confirm the added value of satellite observations in otherwise data-sparse regions (like the Congo catchment in central Africa). However, we also find a number of places where GPCC fits better to GRACE than GPCP which suggests that systematic deficits in satellite observations might also degrade the combined product in certain areas. Despite the differences identified above, we conclude that there is a strong relation between precipitation and storage monitored by satellite gravimetry, implying that GRACE observations should be used more

frequently in the future for large-scale hydrometeorological research.

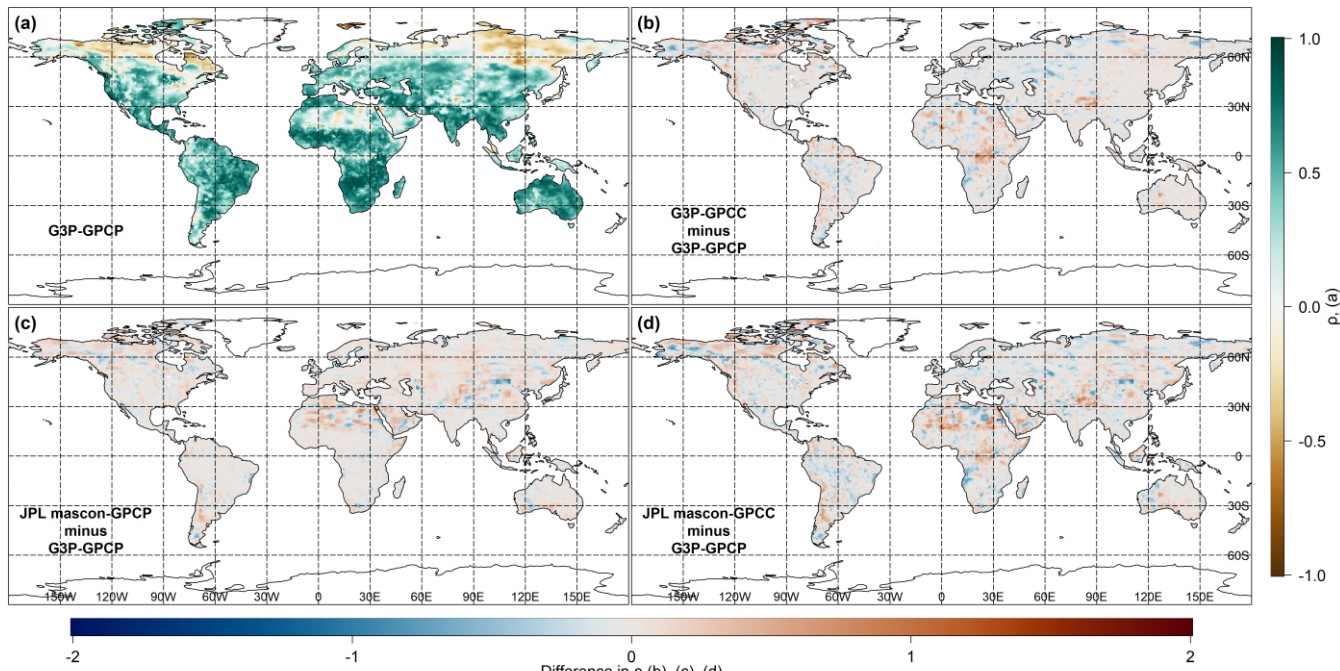

**Figure 2. Display of correlation coefficients between different dTWSA (i.e., G3P, JPL mascon) and cdPA (i.e., GPCP, GPCC) datasets. (a) Correlation coefficients obtained for dTWSA from G3P and cdPA from GPCP (G3P&GPCP). Differences in correlation coefficients relative to G3P&GPCP for (b) G3P&GPCC; (c) JPL mascon&GPCP; and (d) JPL mascon&GPCC. Please note that**
**regions with negative correlations were removed from subsequent analyses.**

Figure 3a illustrates the spatial distribution of $\beta_1$ (Eq. 4), exceeding 0, for G3P&GPCP. We note that certain regions of North Africa, North America, and Northeastern Asia, the $\beta_1$ was less than 0. TWS in these regions were likely influenced by factors other than precipitation, thereby making the link between precipitation and TWSA less reliable. Most regions with polar climates (i.e., Köppen-Geiger Climate Zone E) exhibited $\beta_1$ below zero, indicating a weak link between precipitation anomalies

and changes in TWS in these areas. Similar to the correlation analysis, a contrasting pattern emerged between the arid regions. North America's arid regions (i.e., Zone B) showed a pattern more comparable to Australia's arid regions. Africa's arid regions



had a higher percentage of masked-out areas compared to those in Australia and North America. Additionally, the remaining areas in North Africa's arid regions have lower $\beta_1$ values.

While switching from G3P to the JPL mascon (Figure 3c), we note that $\beta_1$ values are almost same for the global average (mean difference of Figure 3c = -0.01). A quite similar pattern is also emerging when switching from GPCP to GPCC (Figure 3b) and the overall largest decrease in $\beta_1$ values are found for the combination G3P&GPCC (-0.06). In particular, using GPCP in Asia's snow zone and Australia's arid zone revealed more regions with $\beta_1$ closer to 1 than using GPCC. This suggests less need for additional variables to explain the relationship between precipitation anomalies and TWS changes in these regions when

using GPCP than when using GPCC. In Europe's warm temperature zone, the use of the JPL mascon revealed more regions with $\beta_1$ larger than 3 compared to that of G3P.

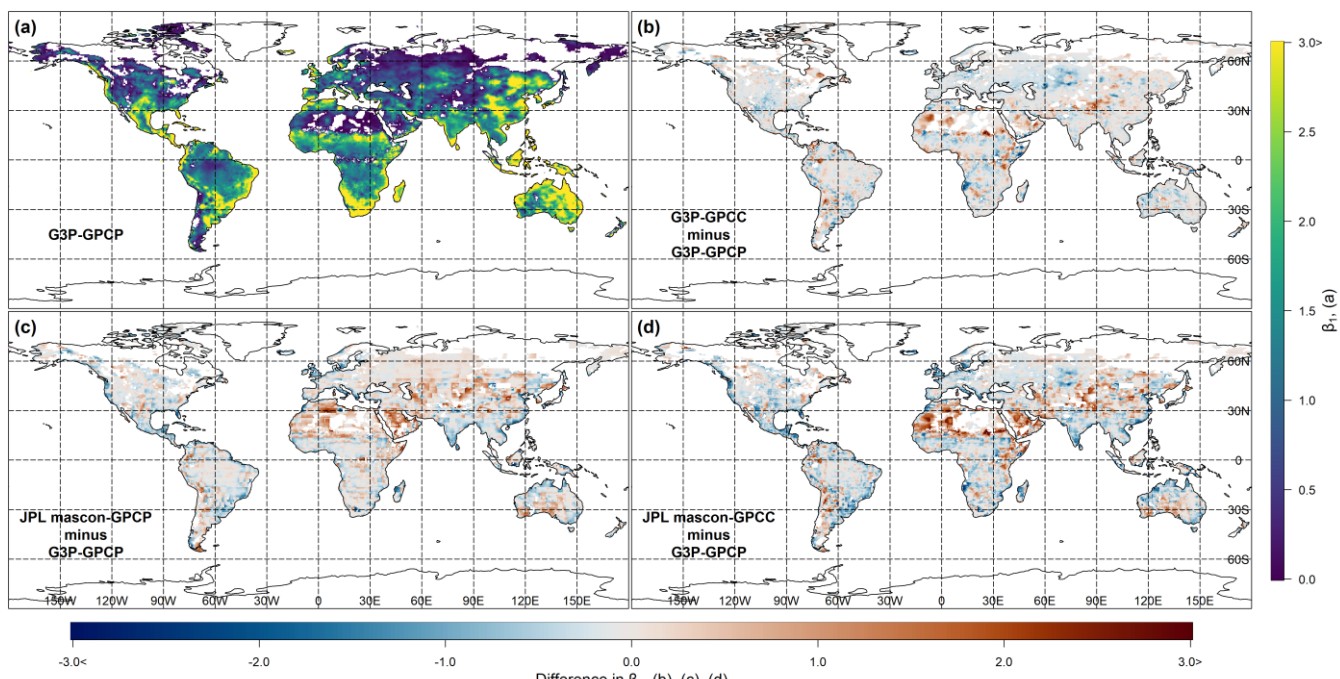

**Figure 3. Display of $\beta_1$ values between different dTWSA (i.e., G3P, JPL mascon) and cdPA (i.e., GPCP, GPCC) datasets. (a) $\beta_1$ values obtained for dTWSA from G3P and cdPA from GPCP (G3P&GPCP). Differences in $\beta_1$ values relative to G3P&GPCP for (b)**
**G3P&GPCC, (c) JPL mascon&GPCP, and (d) JPL mascon&GPCC. Regions where $\beta_1$ value is smaller than 0 are shown in white. Please note that regions with $\beta_1$ value less than 1 were deemed having a weak precipitation-storage relationship and were excluded from subsequent analyses.**

Figure 4a displays a time series of dTWSA derived from both G3P and JPL mascon TWSA datasets in an example region in Australia (133.75° E, 16.75° S). Figure 4b illustrates the time series of cdPA derived from both GPCC and GPCP for the same
region. These visualizations allow us to observe and analyze the fluctuations in water storage deviations and cdPA over time, providing insights into the dynamics of water availability and precipitation, and potential drought recovery patterns. There





existed close agreement between the time series of G3P and JPL mascon, and GPCC and GPCP, as well as between dTWSA and cdPA time series (average ρ = 0.65), as shown in Fig. 4.




**Figure 4. Time series of (a) dTWSA obtained from both G3P and JPL mascon products; and (b) cdPA obtained from both the GPCC and GPCP precipitation products, each in an example region in Australia (133.75° E, 16.75° S).**

**3.2 DRT Estimations**

The spatial distributions of mean DRT estimates based on storage deficit and required precipitation amount from the G3P-
GPCP coupled product are shown in Figures 5a and 6a, respectively. Figures 5b-d and 6b-d depict the spatial distribution of the differences between the mean DRT estimates derived from G3P&GPCP and G3P&GPCC, JPL mascon&GPCP, and JPL mascon&GPCC, respectively, for both methods. The mean DRT estimations based on storage deficit were identical in Figs. 5a and 5b as well as Figs. 5c and 5d because the DRT estimations relied solely on TWS products for this method. Precipitation products were only used for the exclusion of the regions for this method in Fig. 5. Therefore, the only differences between
G3P&GPCP and G3P&GPCC (Fig. 5b) as well as JPL mascon&GPCP (Fig. 5c) and JPL mascon&GPCC (Fig. 5d) were the excluded regions. Even though the required precipitation method incorporated precipitation data into DRT estimations, the



overall spatial patterns of mean DRT remained similar between G3P&GPCP and G3P&GPCC (Fig. 6b) as well as JPL mascon&GPCP and JPL mascon&GPCC. Figures 5 and 6 indicated that both the mean and spatial distributions of DRT estimations were consistent with each other using both methods.

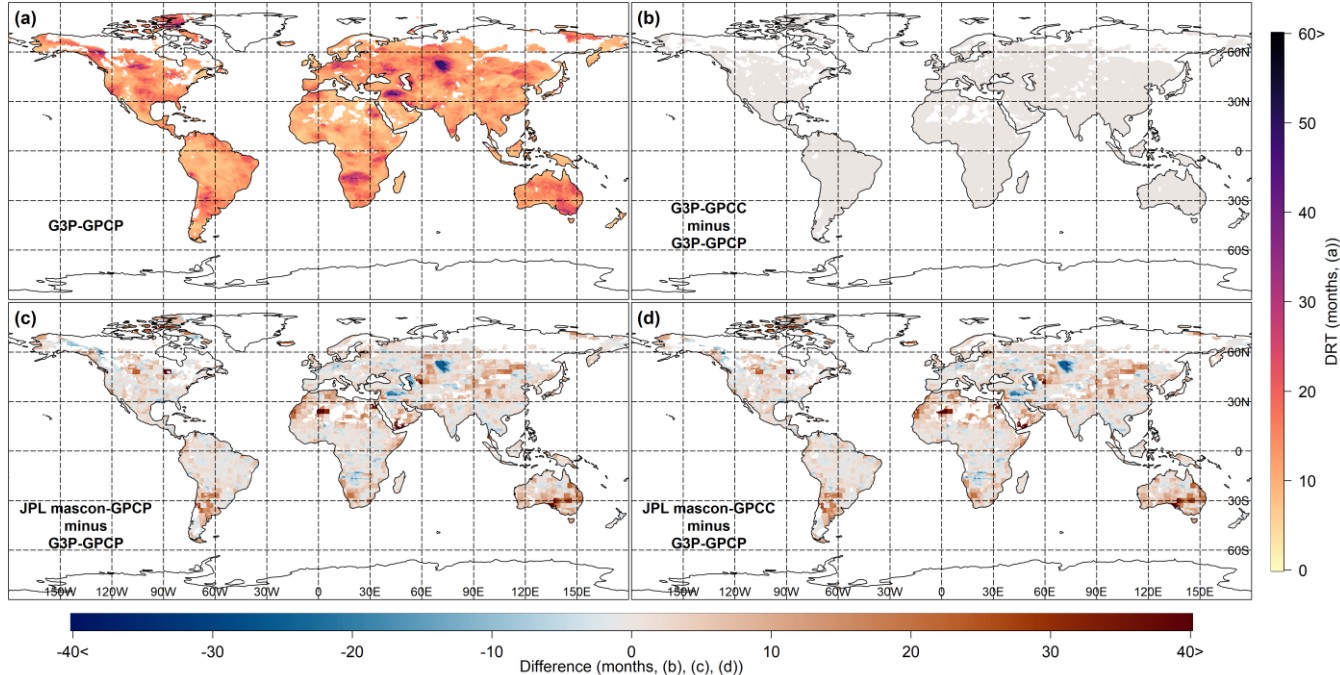


**Figure 5. Display of mean drought recovery time (DRT) estimations based on storage deficit obtained from different dTWSA (i.e., G3P, JPL mascon) and cdPA (i.e., GPCP, GPCC) datasets. (a) DRT obtained for dTWSA from G3P and cdPA from GPCP (G3P&GPCP). Differences in DRT relative to G3P&GPCP for (b) G3P&GPCC, (c) JPL mascon&GPCP, and (d) JPL mascon&GPCC.**



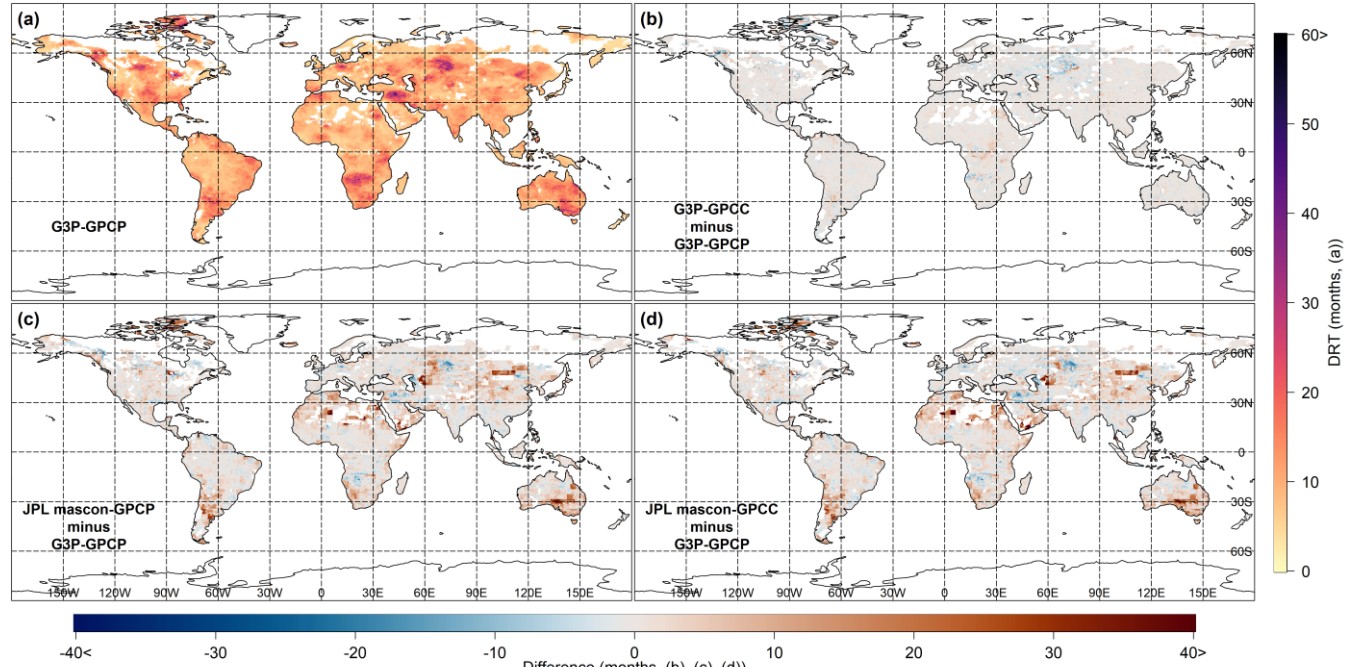

**Figure 6. Display of mean drought recovery time (DRT) estimations based on required precipitation obtained from different dTWSA (i.e., G3P, JPL mascon) and cdPA (i.e., GPCP, GPCC) datasets. (a) DRT obtained for dTWSA from G3P and cdPA from GPCP (G3P&GPCP). Differences in DRT relative to G3P&GPCP for (b) G3P&GPCC, (c) JPL mascon&GPCP, and (d) JPL mascon&GPCC.**

Figures 5a and 5b, which utilized G3P as TWS product with GPCC and GPCP for precipitation, respectively, reveal the highest mean DRT (50-60 months) estimates based on the storage deficit method in Iran and Central Asia. Likewise, for the required precipitation amount method, Figs. 6a and 6b, which also utilized G3P as TWS product with GPCC and GPCP for precipitation, respectively, showed the highest mean DRT in the same regions. Both methods consistently identified Iran, central Asia, southeastern Australia, and northern Africa as the regions experiencing the highest mean DRT (50-60 months), as illustrated in Figs. 5c and 5d , as well as Figs. 6c and 6d, which utilized JPL mascon as the TWS product with GPCC and GPCP for precipitation, respectively. The spatial correlation between Fig. 5a and 6a is 0.75. This shows a high level of spatial correlation between the Fig. 5a and 6a.

The following other regions exhibited high DRT estimates based on both methods and across all the product combinations: central and southern South America (~40 months), central and southern Africa (~45 months), eastern Australia (~35 months), central and western North America (~40 months), central Europe (~35 months), and eastern Asia (~30 months). Increasing global aridity and drought areas since the mid-20th century mainly as a result of extensive drying in eastern Australia and northern mid-latitude regions, as reported by Dai (2011), are consistent with the findings of high DRT in the present study. Eastern Australia (~35 months) experienced more severe drought conditions than western Australia (~20 months) based on





both methods. Consistent with the results of this study, previous research has focused on monitoring droughts in regions with a history of severe, multi-month drought events, such as Iran, central Europe, central and western North America, southeast Australia, and central South America, also, which have experienced higher drought severity compared to other regions (Dai, 2011; Madadgar & Moradkhani, 2014; Rubel et al., 2017; Wu et al., 2023).  Over the Colorado River Basin, Madadgar & Moradkhani (2014) observed droughts of varying severity from 2001 to 2004, lasting a total of 48 months, for the period from

2000 to 2011. Our findings for the same region indicate a mean DRT of approximately 30 months. The results of this study (Figs. 5a and 6a) are consistent with those of Boergens et al. (2020), which found that Central Europe is a drought-prone region, experiencing extreme drought during the consecutive summers of 2018 and 2019, with recovery taking over a year. Moreover, the mean DRT estimations obtained from both methods when using JPL mascon were greater than those when using G3P. As shown in Fig. 6, the close agreement between GPCC and GPCP regarding the spatial distribution of mean DRT

estimates for both TWS products (G3P and JPL mascon) indicates that the choice of precipitation product (GPCC or GPCP) may not influence the overall spatial patterns of DRT estimates. Figures A1 and A2 illustrate the spatial distributions of the SE of DRT estimates, which were similar to the spatial distributions of the mean DRT estimates. Regions with the highest mean DRT also exhibited the highest SE, indicating that those experiencing longer DRT periods showed greater variability in the DRT estimates.


The mean DRT estimations based on storage deficit and required precipitation amount for the Köppen-Geiger main climate zones using all the TWS-precipitation coupled products are shown in Figs. 7a and 7b, respectively. Error bars representing the 95% confidence intervals for each zone indicate variability in the mean DRT estimates. The "n" values show the number of grids per coupled product within each zone. For both methods, the polar (E) zone exhibited the highest mean DRT (18.1

months for storage deficit, 14.2 months for required precipitation amount). Except the polar (E) zone, consistent with previous findings (Van Lanen et al., 2013), for both methods, the arid (B) zone exhibited the highest mean DRT (14.8 months for storage deficit, 12.9 months for required precipitation amount), while the equatorial (A) zone displayed the lowest (10.9 months for storage deficit, 9.7 months for required precipitation amount). Mean DRT based on storage deficit and required precipitation amount were 13.9 months and 11.4 months in the warm temperature (C) zone, respectively, whereas 14.1 months and 10.0

months in the snow (D) zone, respectively. In particular, all the climate zones except the polar (E) zone displayed minimal variability (< 0.2 months) in the mean DRT (indicated by narrow 95% confidence intervals), suggesting low uncertainty. Overall, the difference between G3P and JPL mascon is the highest in the arid (B, 3.8 months) and polar (E, 5.7 months) zones, whereas the differences in the other zones are smaller than the arid (B) and polar (E) zones. Figures A3a and A3b show SE for the DRT estimations based on storage deficit and required precipitation amount, respectively, across the Köppen-Geiger

climate zones for all the TWS-precipitation coupled products. The polar (E) zone had the highest SE for both methods, while the lowest varied by the product. SE for GPCC and GPCP were similar except in the polar (E) zone. JPL mascon estimates had slightly larger SE than G3P for both methods.





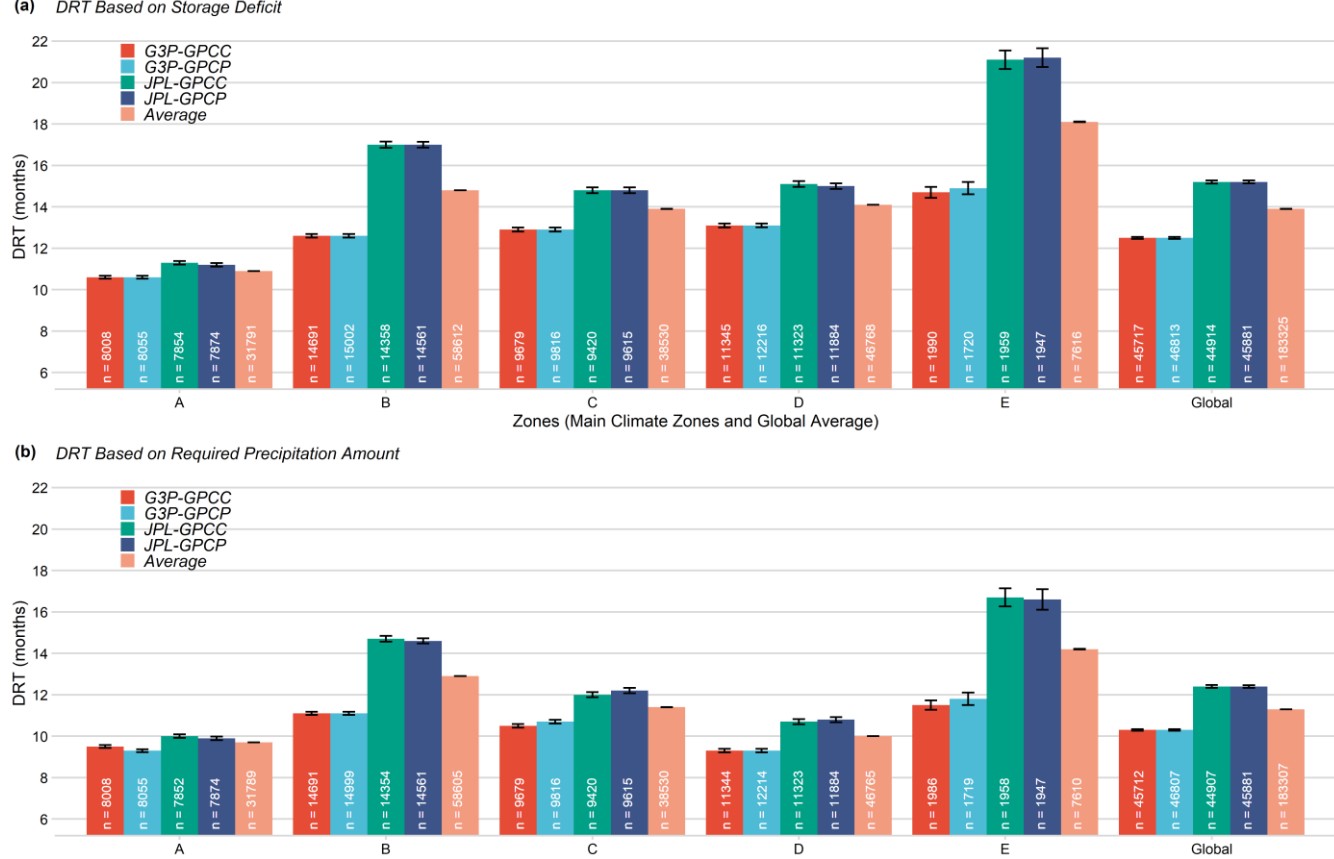

**Figure 7.** Average DRT estimations based on (a) storage deficit and (b) required precipitation amount for various climate zones from two different dTWSA (i.e., G3P and JPL mascon) and two different cdPA (i.e., GPCP and GPCC) products calculated for Equatorial (A); Arid (B); Warm Temperatures (C); Snow (D); and Polar (E) as given by the Köppen-Geiger classification.

On average, DRT based on storage deficit was estimated as 13.9 months, whereas DRT based on required precipitation amount was estimated as 11.3 months. When considering TWS products, regardless of the precipitation products, DRT estimations using JPL mascon (14.2 months) are consistently higher than DRT estimations using G3P (11.6 months), across all climate zones and the global average. When considering precipitation products, DRT estimations using GPCC and GPCP yielded similar values (12.9 months) regardless of the TWS products across all climate zones and the global average. These findings suggest that GPCC and GPCP closely agree on DRT estimations, and the storage deficit derived from G3P was consistently lower than that from JPL mascon across all zones.

## 3.3 Consistency in DRT Estimations

Figure 8a shows the spatial distributions of the consistency categories (Table 1) for the DRT estimates from G3P&GPCP coupled products. Figures 8b-d illustrates the spatial distributions of the differences in consistency categories for the DRT estimations between G3P&GPCP and G3P&GPCC, JPL mascon&GPCP, and JPL mascon&GPCC, respectively. Most regions



fell into consistency category 1 (high agreement) and the mean absolute difference between DRT estimations calculated from both methods is 1.9 months. The spatial patterns were similar across all possible data combinations (Figs. 8a-d), including

those using the different TWS (G3P vs. JPL mascon) and precipitation (GPCC vs. GPCP) products. This consistency was also observed in the mean DRT for all the pairs. As expected, the regions in category 4 (time difference > 9 months) had the highest mean DRT and SE in both methods (Table 2).

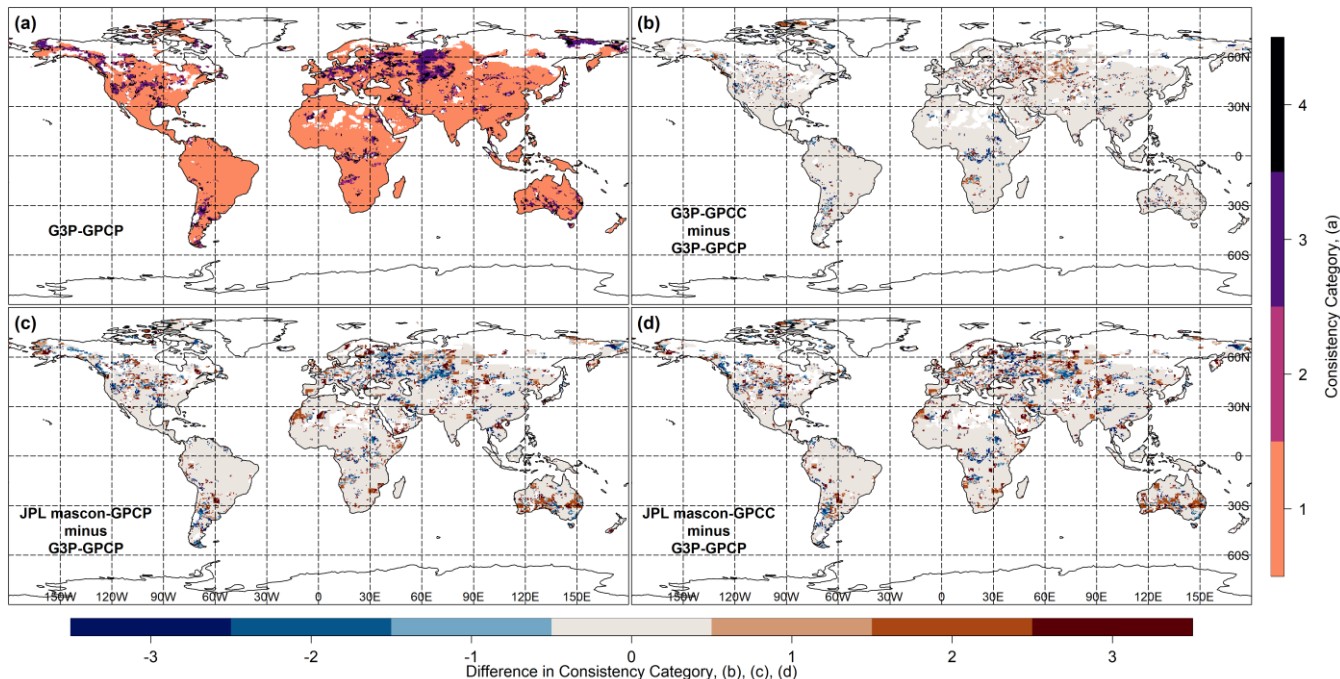

**Figure 8. Display of the consistency in DRT estimations based on either storage deficit or required precipitation obtained from different dTWSA (i.e., G3P, JPL mascon) and cdPA (i.e., GPCP, GPCC) datasets. (a) Consistency for dTWSA from G3P and cdPA from GPCP (G3P&GPCP). Differences in consistency relative to G3P&GPCP for (b) G3P&GPCC, (c) JPL mascon&GPCP, and (d) JPL mascon&GPCC.**

**Table 2. Mean DRT and SE for the Consistency Categories**

|  | Mean DRT (months) | Mean SE (months) |
|---|---|---|
| Category 1 | 12.2 | 3.9 |
| Category 2 | 13.0 | 4.7 |
| Category 3 | 16.0 | 6.9 |
| Category 4 | 23.1 | 12.1 |


Figure 9 shows the consistency levels in terms of the percentage of category 1 (time differences of 1-2 months) for the Köppen-Geiger climate zones using all the coupled products. The polar (E) zone had the lowest average consistency (74.9%), while



the equatorial (A) zone had the highest (97.8%). Overall, 87.5% of DRT estimations achieved category 1 consistency. The consistency rate of G3P (88.5%) is higher than that of JPL mascon (83.5%) across the climate zones and globally. The

G3P&GPCP combination achieved the highest consistency (on average, 88.7%), whereas JPL mascon&GPCP and showed the lowest (on average, 82.8%).

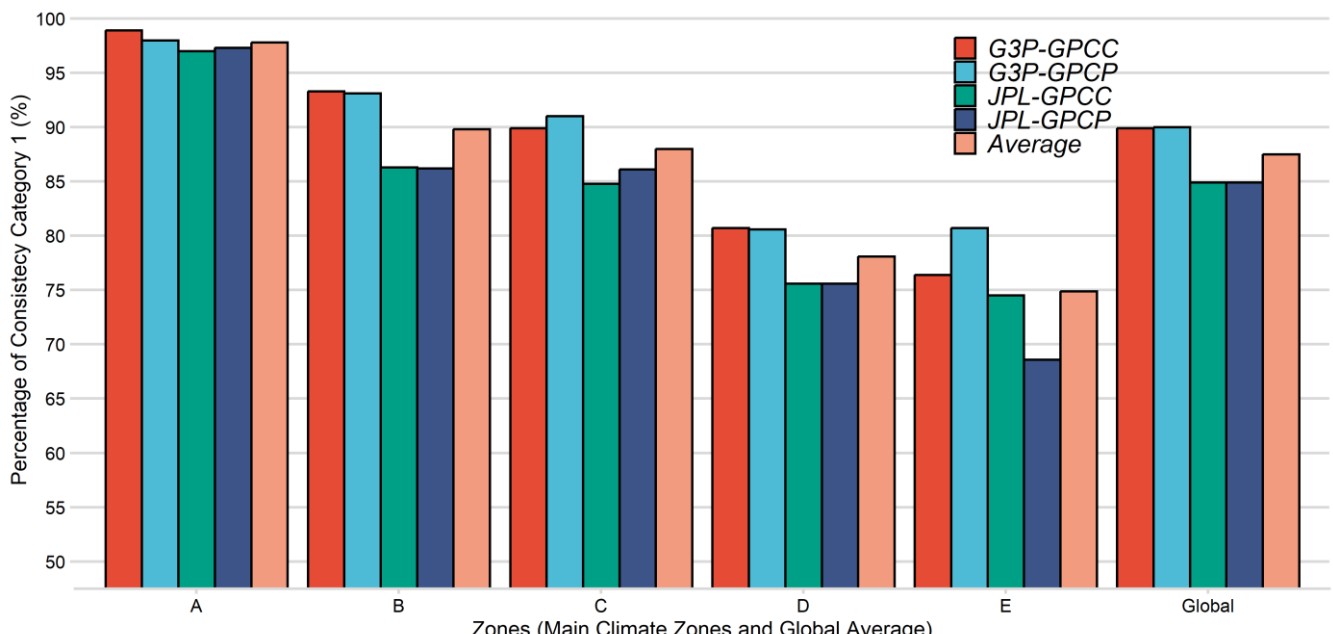

**Figure 9. Percentage of DRT estimations whose consistency is category 1 for different climate zones using all the TWS-precipitation coupled products for climates characterized as Equatorial (A); Arid (B); Warm Temperatures (C); Snow (D); and Polar (E) by the**
**Köppen-Geiger classification.**

The choice of the precipitation product (GPCC vs. GPCP) exerted minimal impact on consistency (average absolute difference 1.4%) when the same TWS product was used. Conversely, G3P led to higher DRT consistency (average absolute difference 5.0%) than JPL mascon, when the same precipitation product was used. The climate zones also influenced consistency. GPCC and GPCP showed similar consistency in the arid (B, 0.2 months) zone and in the snow (D, 0.1 months) zone and the most

different in the polar (E) zone (5.1% difference). In contrast, G3P and JPL mascon had the most similar consistency in the equatorial (A) zone (1.3% difference) and the highest difference in the arid (B) zone and the polar (E) zone (7.0% difference).

## 4. Summary and Conclusions

TWS changes from one time epoch to another as observed by satellite gravimetry are closely related to the precipitation amount that occurred during that time interval. The novel observing concept realized with the GRACE and GRACE-FO missions thus

provides a unique opportunity to evaluate frequently used global precipitation products timescales of a month and longer. Based on our assessments utilizing TWS (G3P spherical harmonics and JPL mascons) and precipitation (GPCC and GPCP)





products, we find a generally high correlation among the two over semi-arid and even wetter climates, some parts of the equatorial, warm temperature and snow zones, with (on global average) the best correspondence between G3P and GPCP. Apart from arid regions with very little storage variability, correlation coefficients are fairly insensitive to the selection of
GRACE products. When switching from GPCP to GPCC, however, we identify changes in correlation in particular in Africa and partly also in Central Asia, where in situ station coverage is poor and the impact of satellite-based precipitation information in GPCP is comparatively large due to the sparse coverage with in situ rain gauges that GPCC solely relies on.

GRACE/GRACE-FO directly provides water storage anomalies, which enables a new way to characterize drought by means
of the storage deficit. The time required to recover from a drought can be thus obtained from the temporal evolution of the storage deficit directly (Singh et al., 2021), so that both duration and severity of a drought can be measured. We find that the mean DRT estimations obtained from either GPCC or GPCP do not significantly differ from one another. However, the mean DRT estimations using JPL mascon were slightly higher, on average 2.6 months, than those using G3P for the global and Köppen-Geiger climate zones. Moreover, the results show that the lowest mean DRT estimation was obtained in the equatorial
(A) zone, whereas the highest was in the polar (E) zone for all TWS-precipitation combinations. Conversely, G3P showed slightly higher consistency (5.0%) in the DRT estimations than JPL mascon when the same precipitation product was used. Additionally, the results indicated that the equatorial (A) zone showed the highest consistency in mean DRT estimations, whereas the polar (E) zone showed the lowest consistency in mean DRT estimations for all TWS-precipitation pairs under consideration.

The results of our study underline the potential value of GRACE for hydrometeorologic research due to the tight relationship between precipitation and TWS changes. Its global coverage (with albeit rather low spatial resolution) allows the testing of different precipitation products not only from combinations of varying satellite and in situ observations (as performed in this study) but also from numerical weather prediction models and global atmospheric reanalyses. Both NASA and also the
European Space Agency (ESA) are currently working on future satellite gravity missions with even more precise sensors, so that further improvements in the quality of satellite gravimetry products for hydrologic applications can be expected.



**Appendix A**

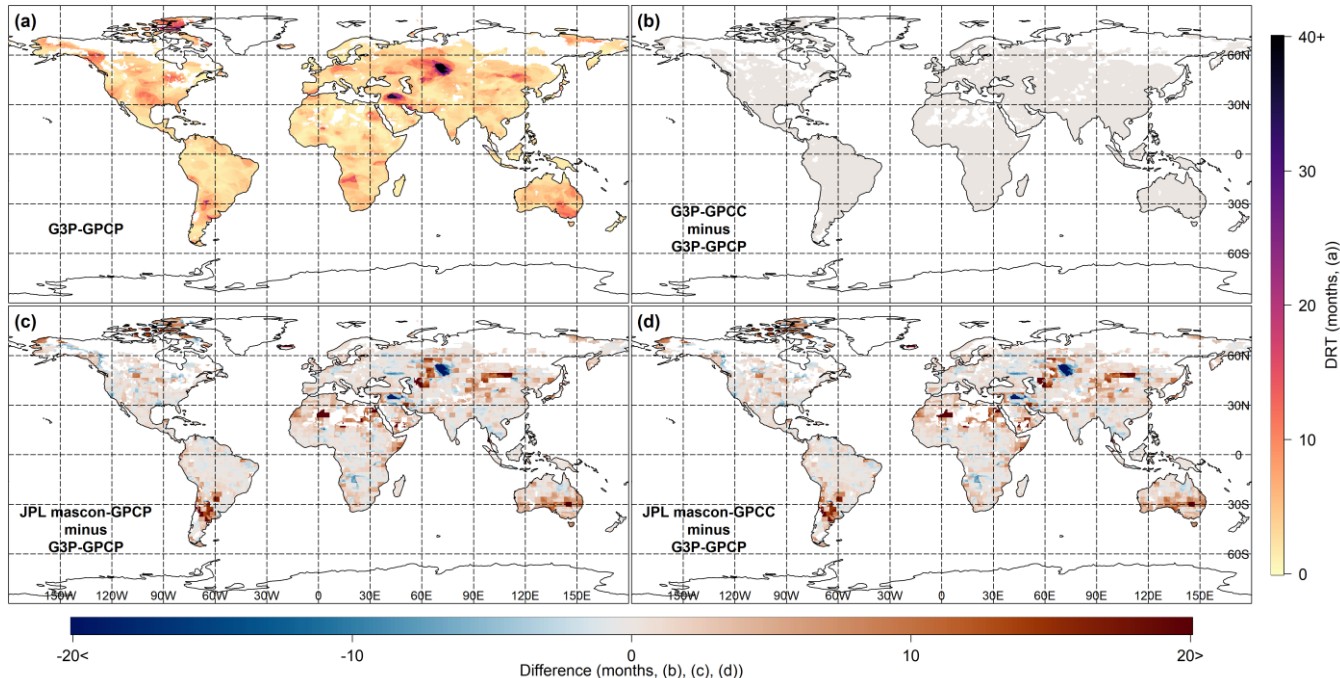

**Figure A1. Display of the standard error of DRT estimations based on the storage deficit obtained from different dTWSA (i.e., G3P, JPL mascon) and cdPA (i.e., GPCP, GPCC) datasets. (a) Standard error for dTWSA from G3P and cdPA from GPCP (G3P&GPCP). Differences in standard error relative to G3P&GPCP for (b) G3P&GPCC, (c) JPL mascon&GPCP, and (d) JPL mascon&GPCC.**



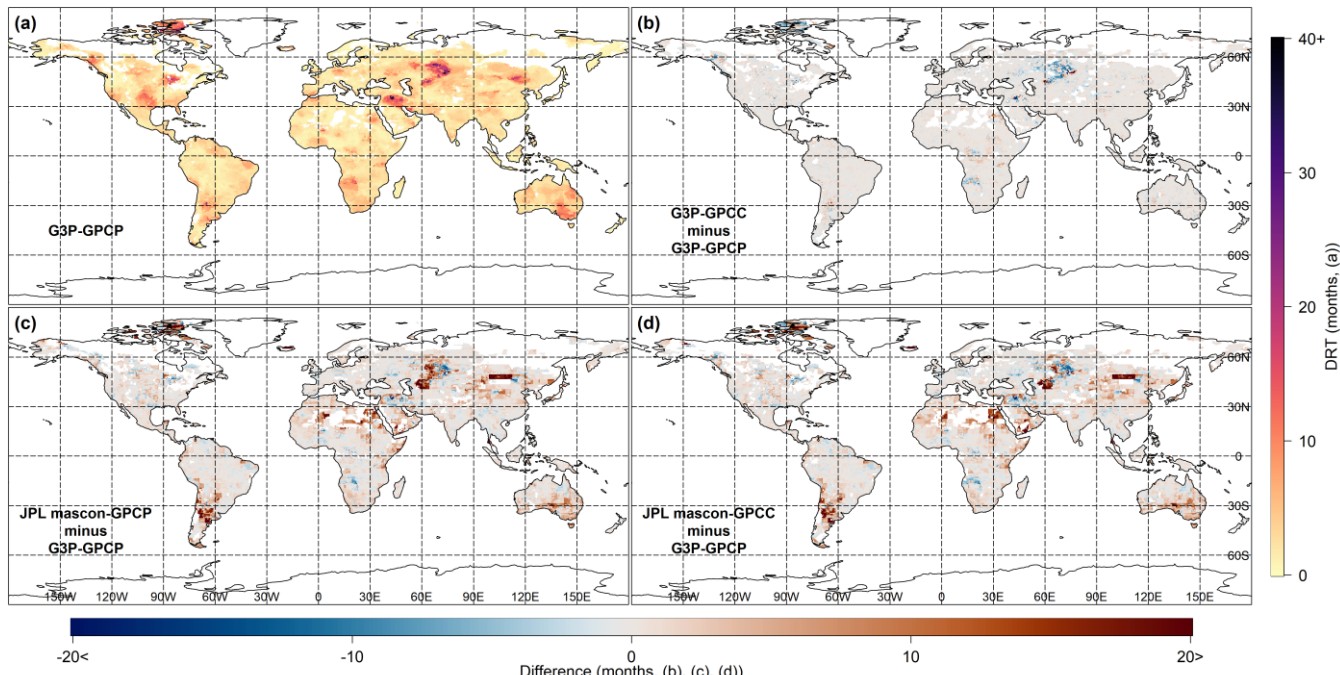

**Figure A2. Display of the standard error of DRT estimations based on the required precipitation obtained from different dTWSA (i.e., G3P, JPL mascon) and cdPA (i.e., GPCP, GPCC) datasets. (a) Standard error for dTWSA from G3P and cdPA from GPCP (G3P&GPCP). Differences in standard error relative to G3P&GPCP for (b) G3P&GPCC, (c) JPL mascon&GPCP, and (d) JPL mascon&GPCC.**



**Figure A3. Standard error for average DRT estimations based on (a) storage deficit and (b) required precipitation amount for various climate zones from two different dTWSA (i.e., G3P and JPL mascon) and two different cdPA (i.e., GPCP and GPCC) products calculated for Equatorial (A); Arid (B); Warm Temperatures (C); Snow (D); and Polar (E) as given by the Köppen-Geiger classification.**

## Data Availability

The datasets used for this study are publicly available under the following links. JPL mascon GRACE and GRACE-FO TWS: https://cmr.earthdata.nasa.gov/virtual-directory/collections/C2536962485-POCLOUD/temporal/2002/04/16; G3P GRACE and GRACE-FO TWS: ftp://isdcftp.gfz-potsdam.de; GPCC Full Data Monthly precipitation: https://opendata.dwd.de/climate_environment/GPCC/html/fulldata-monthly_v2022_doi_download.html; GPCP v3.2 Satellite-Gauge (SG) Combined Data: https://disc.gsfc.nasa.gov/datasets/GPCPMON_3.2/summary; Köppen-Geiger climate classification scheme: https://koeppen-geiger.vu-wien.ac.at/present.htm.



**Author Contribution**

ÇÇ contributed to conceptualization, data curation, formal analysis, investigation, methodology, resources, software, validation, visualization, preparing the original draft, and editing the manuscript according to the reviews from the co-authors. MTY contributed to funding acquisition, project administration, supervision, resources, and reviewing and editing the manuscript. HD contributed to conceptualization, methodology, supervision, and reviewing and editing the manuscript. ESI contributed to funding acquisition, project administration, supervision, and reviewing and editing the manuscript. FE, CF, and ALY contributed to funding acquisition, and reviewing and editing the manuscript.

**Competing Interests**

The authors declare that they have no conflict of interest.

**Acknowledgments**

We thank the German Research Center for Geosciences (GFZ) and Jet Propulsion Laboratory (JPL) for providing GRACE and GRACE-FO datasets; Deutscher Wetterdienst (DWD) for providing the GPCC Full Data Monthly Product; Goddard Earth Sciences Data and Information Services Center (GES DISC) for providing GPCP version 3.2 Satellite-Gauge Combined Precipitation dataset; The University of Veterinary Medicine, Vienna for providing Updated World Map of the Köppen-Geiger Climate Classification. This study is funded by The Scientific and Technological Research Council of Turkey (TUBITAK) Grant Number 120R065 and the Federal Ministry of Education and Research (BMBF) with the Grant Number 01DL22002.

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
