# Peer review of "Evaluation of Globally Gridded Precipitation Data and Satellite-Based Terrestrial Water Storage Products Using Hydrological Drought Recovery Time"

_EGUsphere, 2024_

## Referee Comment (RC2)

---------------------- **REVIEW REPORT**---------------------

TITLE: Use of GPCC and GPCP Precipitation Products and GRACE and GRACE-FO Terrestrial Water Storage Observations for the Assessment of Drought Recovery Times
AUTHORS: Çağatay Çakan, M. Tuğrul Yılmaz, Henryk Dobslaw, E. Sinem Ince, Fatih Evrendilek, Christoph Förste, Ali Levent Yagci

Rating at 1 to 4 scale, 1 means excellent and 4 means poor score:

----------- Scientific significance -----------
SCORE: 2 (Good)

----------- Scientific quality-----------
SCORE: 2 (Good)

---------- Presentation quality-----------
SCORE: 2 (Good)

----------- Overall Rating -----------
SCORE: 2 (Good)

----------- Guidance -----------

**This paper is accepted to subject a minor revision**:

The study is well-organized and innovative, with its originality stemming from the use of GRACE and GRACE-FO terrestrial water storage (TWS) data as an independent approach for evaluating the accuracy of precipitation products. The authors computed Drought Recovery Times (DRT) from Total Water Storage Anomaly (TWSA) data using two different approaches. The first approach, referred to as the "storage deficit" method, relies solely on TWSA data, while the second, the "required precipitation amount" method, integrates TWSA with precipitation data. Two TWSA products, JPL and G3P, were utilized for these calculations. Additionally, the authors evaluated the discrepancies in DRT results between the two TWSA products. I have only a few minor comments for the authors may consider:

**Minor Comments:**

- Line 285 : Which correlation method did you use, Can you please name it? (e.g. Pearson's correlation)
- Line 298 : Please verify the figure numbers, as they might need to be labeled as 2b and 2d
- The paper uses numerous abbreviations and technical terms, so it is recommended to include a glossary of full forms for the abbreviations after the conclusion.

Overall, this research is well-structured and presents a state-of-the-art contribution to hydrology by highlighting the potential of GRACE and GRACE-FO data in evaluating precipitation products and drought characteristics. Addressing the minor comments provided will further strengthen the manuscript's impact.

---

## Author Comment (AC1)

**Referee #1**

**1. Overview**

The article proposes using Terrestrial Water Storage Anomalies (TWSA) from GRACE missions to independently evaluate precipitation products, specifically GPCC and GPCP, since traditional methods rely on gauge data used in the products' creation. By calculating Drought Recovery Times (DRT) with TWSA alone and combined with precipitation data, the authors aim to provide a more robust assessment of the alignment between precipitation products and TWSA.

Authors: We thank the reviewer for the invaluable comments. In this version, we have revised the text according to the suggestions. Our responses are given below in red while the reviewer's comments are given in black.

**2. Major comments**

Throughout the article, I encountered many interesting results, but the core scientific analysis was lacking. It was unclear what these findings reveal, how they answer the research question, or how the new method clarifies the strengths and weaknesses of the precipitation products. Specifically, there is no discussion of contexts in which one product outperforms the other, nor an exploration of why this might be.

Additionally, the research question itself is not well-defined. While the abstract claims the goal is to evaluate precipitation products using TWSA data to calculate DRT, the paper sometimes shifts focus to evaluating TWSA products or comparing which precipitation/TWSA combination best estimates DRT.

To address this, I recommend: 1) clearly defining the research question in the introduction and maintaining alignment throughout, and 2) adding a discussion section (included in or separate from the " results " section) to interpret the results in light of the research question.

Authors: Thank you for your helpful comments. In this revised version, we have substantially refined the research question and integrated a discussion section within the results. We have also addressed the other feedback received. To clarify the research question, we have updated the abstract introduction and added a discussion section as suggested as follows:

*"This study aims to assess the accuracy of the Global Precipitation Climatology Center (GPCC) Full Data Monthly Product v2022 and Global Precipitation Climatology Project (GPCP) v3.2 Monthly Analysis Product by estimating hydrological drought recovery time (DRT) from precipitation and terrestrial water storage anomaly (TWSA) acquired from satellite gravimetry. This study also evaluates the performance of G3P and JPL mascon TWS monthly-solutions from the Gravity Recovery and Climate Experiment (GRACE) and GRACE Follow-On (GRACE-FO) satellite missions."*

Following the above comment, we now revise the title of the manuscript to better reflect the focus of the study.

*"Evaluation of Globally Gridded Precipitation Data and Satellite-Based Terrestrial Water Storage Products Using Hydrological Drought Recovery Time"*

Additionally, to better highlight the goal of the study, now the relevant section in introduction is modified as follows:

*"The current study aims to independently evaluate and compare frequently used global gridded precipitation products (i.e., from GPCC and GPCP) by using the GRACE/GRACE-FO TWS data (i.e., the JPL mascon and G3P products) in order to assess drought conditions. Also, this research evaluates the performance of the JPL mascon and G3P TWS products. Both evaluation is conducted by estimating DRT based on TWSA and required precipitation amount. Comparing the suitability of these precipitation and TWS products for global hydrological applications across various Köppen-Geiger climate zones enhances our understanding of the relationship between hydrological droughts and global precipitation and TWS products through DRT estimations."*

Again, following the reviewer's comments, we have now added a new section called *3.4 Discussions* to interpret the results in light of the research question, as follows:

*"Both precipitation products provided similar global mean DRT estimations (12.6 months), with a high consistency rate of 87.5%. The largest discrepancy in mean DRT estimation (0.1 months) was observed in the polar (E) zone, with no significant difference in the snow (D) zone. The consistency between the two precipitation products was less than 1% across all the climate zones.*

*For the TWS products, the global mean DRT estimation using JPL mascon (13.8 months) was 2.6 months higher than that of G3P (11.4 months). G3P exhibited 5.0% higher global consistency (90.0%) than did JPL mascon (85.0%). The largest difference in mean DRT estimation between G3P (13.2 months) and JPL mascon (18.9 months) occurred in the polar (E) zone, with the smallest difference between G3P (10.0 months) and JPL mascon (10.6 months) in the equatorial (A) zone. The greatest consistency disparity (7.0%) occurred in the polar (E) and arid (B) zones, with G3P (78.6% and 93.3%) outperforming JPL mascon values (71.6% and 86.3%) in the polar and arid zones, respectively. The smallest difference (1.3%) between G3P (98.5%) and JPL mascon (97.2%) was recorded in the equatorial (A) zone. In terms of consistency across all the climate zones, G3P outperformed JPL mascon."*

Finally, the second paragraph of section *4 Summary and Conclusions* have been substantially revised following this comment:

*"GRACE/GRACE-FO directly provide water storage anomalies, offering a novel approach to characterize drought by assessing the storage deficits. The time required for drought recovery can be directly derived from the temporal evolution of these deficits (Singh et al., 2021), enabling the measurement of both drought duration and severity. Both GPCC and GPCP products exhibited not only similar mean DRT estimations but also comparable consistency in their DRT estimations, globally and across all the Köppen-Geiger climate zones. For the TWS products, the mean DRT estimations from JPL mascon were, on average, 2.6 months higher, than those from G3P, globally and across all the Köppen-Geiger climate zones. However, G3P*

*showed slightly higher consistency in the DRT estimations (5.0% difference) than did JPL mascon. Furthermore, G3P demonstrated greater consistency than JPL mascon across all the Köppen-Geiger climate zones."*

**3. Minor comments**

**Abstract**: Consider simplifying the abstract by emphasizing the key findings, rather than delving into specific details. This will help focus the reader's attention on the main outcomes without overwhelming them with too much information.

Authors: Now we have simplified the abstract by emphasizing the key findings and also reduced the length from 362 words to 257 words. Revised abstract reads as follows:

*"Accurate precipitation observations are crucial for understanding meteorological and hydrological processes. Most precipitation products rely on station-based observations, either directly or for bias-correcting satellite retrievals. To validate these station-based precipitation products, additional independent data sources are necessary. This study aims to assess the performance of the Global Precipitation Climatology Center (GPCC) Full Data Monthly Product v2022 and Global Precipitation Climatology Project (GPCP) v3.2 Monthly Analysis Product by estimating hydrological drought recovery time (DRT) from precipitation and terrestrial water storage anomaly (TWSA) acquired from satellite gravimetry. This study also evaluates the performance of G3P and JPL mascon TWS monthly-solutions from the Gravity Recovery and Climate Experiment (GRACE) and GRACE Follow-On (GRACE-FO) satellite missions. The current study employed two methods to estimate DRT and evaluated the consistency of DRT estimations by calculating the time difference between the two methods. No significant differences in the mean DRT estimations using GPCC and GPCP were found for both globally and across all climate zones, with comparable consistencies from GPCC and GPCP. For the TWS products, DRT estimation using JPL mascon was, on average, 2.6 months longer than that using G3P. However, the G3P showed approximately 5.0% higher consistency than the JPL mascon globally and across each climate zone. These results indicated a close agreement between GPCC and GPCP in DRT estimations. G3P showed greater consistency in DRT estimation with the precipitation products than JPL mascon. These findings provide valuable insights into the accuracy of precipitation and TWS anomaly products by utilizing hydrological drought characteristics, enhancing our understanding of meteorological and hydrological processes."*

**Line 28**: You present the consistency results without explaining what they entail. Providing context here will help the reader understand the significance of these results.

Authors: To clarify the consistency result, we have added the following text in the manuscript (Abstract):

*"The current study employed two methods to estimate DRT and evaluated the consistency of DRT estimations by calculating the time difference between the two methods."*

**Line 178**: The detrending process would benefit from more explanation. Please consider adding details on the method used, along with a relevant reference.

Authors: To clarify the detrending processes, we have now revised the following text and added the relevant reference in the manuscript. Also, we have added the equation for the detrending process (Section 2.3):

*"To isolate the impact of such long-term processes, we detrended the TWSA data for each grid by removing the linear trend of relevant grid (Singh et al., 2021)."*

$$dTWSA_{x,y,t} = sTWSA_{x,y,t} - trend(sTWSA_{x,y,t}), \qquad\qquad (2)$$

*where $sTWSA_{x,y,t}$ is the smoothed TWSA at x,y grid point and time t, and $trend(sTWSA_{x,y,t})$, is the trend of the smoothed TWSA at x,y grid point and time t."*

**Line 287**: You describe the correlations over Australia (0.55), South America (0.46), and South Africa ($\rho > 0.47$) as "high." Please clarify the criteria or thresholds you used to define these correlations as high.

Authors: All the r values $\leq 0.13$ were not significant ($p > 0.05$; $n = 216$). To clarify the threshold which we used to define these correlations, we provide the following classification in the manuscript (Section 2.5):

*"We classified the r values as follows: no or insignificant correlation (0.0–0.13), weak correlation (0.14–0.39), moderate correlation (0.40–0.69), and strong correlation (0.70–1.0)."*

Also, we have revised the text in *Section 3.1* accordingly:

*"All the r values $\leq 0.13$ were not significant (p > 0.05; n = 216). Significant and moderate correlations were found over Australia (0.55), South America (0.46), and southern Africa (0.60), where not only are water storage variations substantial, but also in situ observing networks are dense. These correlations indicate substantial agreement in these areas."*

**Line 298**: The text references Figures 2c and 2d, but these should be Figures 2b and 2d. Please adjust for accuracy.

Authors: Thank you for your invaluable comment. The references should be Figures 2c and 2d. However, to improve the clarity of the relevant section, we have now revised the text in the manuscript (Section 3.1) as follows:

*"GPCC (Fig. 2d, JPL mascon&GPCC) affected correlations to a larger extent than GPCP (Fig. 2c, JPL mascon&GPCP), in particular over places with less dense in situ networks. Given the standard deviation values of correlation differences (Figs. 2c and 2d) due to switching from GPCP to GPCC, the variability was higher in Fig. 2d (global average: 0.21) than in Fig. 2c (global average: 0.14)."*

---

## Author Comment (AC2)

**Referee #2**

The study is well-organized and innovative, with its originality stemming from the use of GRACE and GRACE-FO terrestrial water storage (TWS) data as an independent approach for evaluating the accuracy of precipitation products. The authors computed Drought Recovery Times (DRT) from Total Water Storage Anomaly (TWSA) data using two different approaches. The first approach, referred to as the "storage deficit" method, relies solely on TWSA data, while the second, the "required precipitation amount" method, integrates TWSA with precipitation data. Two TWSA products, JPL and G3P, were utilized for these calculations. Additionally, the authors evaluated the discrepancies in DRT results between the two TWSA products. I have only a few minor comments for the authors may consider:

Authors: Thank you for your positive feedback. In this version, we have revised the text by addressing your comments. Our responses are given below in red while the reviewer's comments are given in black.

**Minor Comments:**
-Line 285: Which correlation method did you use, Can you please name it? (e.g. Pearson's correlation)

Authors: We used Pearson's correlation coefficient. To clarify this point, we have now revised the following text in the manuscript (Section 2.5):

*"Following the study of Singh et al. (2021), we estimated not only regression coefficients (i.e., β0 and β1) but also the Pearson's correlation coefficient (r) between cdPA and dTWSA, as well as maximum drought length for each pixel utilizing 19 years of monthly data spanning from 2002 to 2020. We classified the r values as follows: no or insignificant correlation (0.0–0.13), weak correlation (0.14–0.39), moderate correlation (0.40–0.69), and strong correlation (0.70–1.0)."*

-Line 298: Please verify the figure numbers, as they might need to be labeled as 2b and 2d

Authors: Thank you for your comment. The references should be Figures 2c and 2d. However, to improve the clarity of the relevant section, we have now revised the text in the manuscript (Section 3.1) as follows:

*"GPCC (Fig. 2d, JPL mascon&GPCC) affected correlations to a larger extent than GPCP (Fig. 2c, JPL mascon&GPCP), in particular over places with less dense in situ networks. Given the standard deviation values of correlation differences (Figs. 2c and 2d) due to switching from GPCP to GPCC, the variability was higher in Fig. 2d (global average: 0.21) than in Fig. 2c (global average: 0.14)."*

-The paper uses numerous abbreviations and technical terms, so it is recommended to include a glossary of full forms for the abbreviations after the conclusion.

Authors: To clarify the abbreviations, we have added the full forms glossary for the abbreviations as a table in the manuscript (Appendix A):

"

**Table A1. The full forms glossary for the abbreviations**

| | |
|---|---|
| cdPA | Cumulative Detrended Precipitation Anomaly |
| cPA | Cumulative Precipitation Anomaly |
| DRT | Drought Recovery Time |
| dTWSA | Deviation of Storage |
| G3P | Global Gravity-based Groundwater Project |
| GPCC | Global Precipitation Climatology Center |
| GPCC FDM | Global Precipitation Climatology Center Full Data Monthly Product |
| GPCP | Global Precipitation Climatology Project |
| JPL mascons | Jet Propulsion Laboratory Mass Concentration blocks |
| scPA | Smoothed Cumulative Precipitation Anomaly |
| TWS | Terrestrial Water Storage |
| TWSA | Terrestrial Water Storage Anomaly |

"

Overall, this research is well-structured and presents a state-of-the-art contribution to hydrology by highlighting the potential of GRACE and GRACE-FO data in evaluating precipitation products and drought characteristics. Addressing the minor comments provided will further strengthen the manuscript's impact.

Authors: Thank you for your encouraging words.

---

## Referee Report (RR1)

**Review of the article: "Evaluation of Globally Gridded Precipitation Data and Satellite-Based Terrestrial Water Storage Products Using Hydrological Drought Recovery Time" by Çakan et al.**

**1. Overview**

I appreciate the efforts made to better outline the objectives in the abstract, introduction, and conclusion, as well as the proposed discussion section. The objectives are more clearly defined, which makes it easier to follow the structure and flow of the article. However, the manuscript still lacks a robust discussion and does not sufficiently highlight the key insights of the study.

**2. Major comments**

**Abstract:**

The abstract would benefit from more information on the implications of the analysis, particularly which product performs better and in which contexts. Highlight the key insights more explicitly to provide a clearer summary of the manuscript's findings.

**Discussion:**

The section titled 3.4 "Discussion" has been included within section 3 "Results." A discussion cannot be a subsection of the results. I recommend renaming section 3 to "Results and Discussion."

The current 3.4 "Discussion" does not provide a proper discussion but rather a detailed description of the results. While this description is interesting and useful, it lacks a synthesis and a discussion of the main findings.

The discussion should address key questions such as for example:

- What do the analyses reveal overall?
- Which precipitation product performs better, and why?
- Which TWSA product performs better, and why?
  - For example, what are the implications of JPL Mascon showing, on average, a DRT that is 2.6 months longer than that derived using G3P?

Additionally, the discussion should address potential limitations of the study to provide a more balanced evaluation.

**Conclusion:**

The manuscript appears to provide an innovative and interesting method for evaluating precipitation products but neglects the main conclusions and their implications regarding the evaluation of GPCC, GPCP, G3P, and JPL Mascon. The conclusion should explicitly state which products show the best performance and in which contexts, along with the broader implications of these findings.

**3. Minor comments**

**Line 24**: The term "performance" is too vague. Please specify what aspect of performance is being referred to.

**Line 27**: The phrase "difference between the two methods" is too vague. Provide a concise explanation of what the difference entails.

**Figure 5**: Avoid showing panels (b) and (d) if there are no precipitation data used in the DRT estimation and they have no impact on the results.

**Figure 8**: If the DRT values presented in this figure are derived as an average between DRT calculated from storage deficit and DRT calculated from required precipitation, this should be clarified explicitly in the caption.

---

## Referee Report (RR2)

**Review of the article: "Evaluation of Globally Gridded Precipitation Data and Satellite-Based Terrestrial Water Storage Products Using Hydrological Drought Recovery Time" by Çakan et al.**

I thank the authors for their thoughtful revisions and clarifications. The objectives, results, and conclusions are now more clearly presented, and the addition of a discussion section helps better contextualize the findings. The manuscript is substantially improved.

However, before recommending acceptance, I suggest a minor revision to improve the discussion of the method's limitations.

The current discussion of limitations remains somewhat superficial. For example, the authors mention line 496 that external factors, such as anthropogenic water use or land cover changes, could influence DRT estimates, and that uncertainties in data retrieval and processing exist. While these points are valid, they are presented quite briefly. A more reflective discussion of these limitations, including potential impacts on the linearity assumption and the overall reliability of DRT estimation, would provide a deeper understanding of the method's scope and constraints.

In particular, it would be helpful to briefly address:

- The role of unobserved hydrological fluxes, such as evapotranspiration and runoff, and how they might affect the relationship between precipitation and storage variations.

- The spatial scale mismatch: while precipitation data are aggregated to the GRACE resolution, a brief discussion of how differences in signal characteristics may still affect comparability would be useful.

- Regression behavior: The authors present a global map of $\beta_1$ values, which is informative, but further clarification is needed on how to interpret high values (e.g., >2 or >3). High $\beta_1$ values could indicate non-linearity in the precipitation-storage relationship or potential methodological issues, such as data noise or model misfit. A brief discussion on this would help assess the model's reliability and its applicability to different regions.

I am not requesting any new analysis, only a more explicit discussion of these points in the text.

---

## Author Response (AR2)

**Referee #1**

**1. Overview**

I appreciate the efforts made to better outline the objectives in the abstract, introduction, and conclusion, as well as the proposed discussion section. The objectives are more clearly defined, which makes it easier to follow the structure and flow of the article. However, the manuscript still lacks a robust discussion and does not sufficiently highlight the key insights of the study.

Authors: We thank the reviewer for the constructive comments. In this revised version, we have substantially strengthened the discussion and better highlighted the key findings of the study. Additionally, we have revised the text in accordance with the other comments.

**2. Major Comments**

**Abstract:**

The abstract would benefit from more information on the implications of the analysis, particularly which product performs better and in which contexts. Highlight the key insights more explicitly to provide a clearer summary of the manuscript's findings.

Authors: We thank the reviewer for this valuable feedback. In this revised version, we have expanded the abstract to explicitly highlight the key findings of the study. We have also included a clearer discussion of the implications of our analysis, specifying which product performs better and under which contexts. We have revised the abstract as follows:

*"Accurate precipitation observations are crucial for understanding meteorological and hydrological processes. Most precipitation products rely on station based observations, either directly or for bias corrected satellite retrievals. To validate these station-based precipitation products, additional independent data sources are necessary. This study aims to assess the performance of the Global Precipitation Climatology Center (GPCC) Full Data Monthly Product v2022 and Global Precipitation Climatology Project (GPCP) v3.2 Monthly Analysis Product by estimating the hydrological drought recovery time (DRT) from precipitation and the terrestrial water storage anomaly (TWSA) acquired from satellite gravimetry. This study also evaluates the drought monitoring performance of G3P and JPL mascon Total Water Storage (TWS) monthly solutions from the Gravity Recovery and Climate Experiment (GRACE) and GRACE Follow-On (GRACE-FO) satellite missions. The current study employed two methods to estimate DRT and evaluated the consistency of DRT estimates by calculating the time difference in DRT values derived from the two methods. Globally and across all climate zones, GPCC and GPCP showed comparable performance in hydrological applications with no significant differences in the mean DRT estimates. For the TWS products, DRT estimates using JPL Mascon were, on average, 2.6 months longer than those using G3P. However, the G3P showed approximately 5.0% higher consistency than the JPL mascon globally and across each climate zone, suggesting its better suitability for more precise drought related analyses. These findings indicate that G3P outperforms JPL Mascon in aligning with precipitation products and offers better consistency in DRT estimation. These results provide valuable insight into accuracy of precipitation and TWSA products by utilizing hydrological drought characteristics, enhancing our understanding of meteorological and hydrological processes."*

**Discussion:**

The section titled 3.4 "Discussion" has been included within section 3 "Results." A discussion cannot be a subsection of the results. I recommend renaming section 3 to "Results and Discussion."

The current 3.4 "Discussion" does not provide a proper discussion but rather a detailed description of the results. While this description is interesting and useful, it lacks a synthesis and a discussion of the main findings.

The discussion should address key questions such as for example:

- What do the analyses reveal overall?

- Which precipitation product performs better, and why?

- Which TWSA product performs better, and why?

  ◦ For example, what are the implications of JPL Mascon showing, on average, a DRT that is 2.6 months longer than that derived using G3P?

Additionally, the discussion should address potential limitations of the study to provide a more balanced evaluation.

Authors: Following the recommendation of the reviewer, now we have renamed Section 3 to 'Results and Discussion' to better reflect the content. Additionally, we have substantially revised the discussion to provide a more comprehensive synthesis of the main findings. Specifically, we now explicitly address key questions regarding the overall implications of the analyses, the relative performance of the precipitation and TWSA products, and the reasons behind their differences. Furthermore, we have incorporated a discussion on the potential limitations of the study to ensure a more balanced evaluation. We have revised 3.4 "Discussion" as follows:

*"Both precipitation products provided similar global mean DRT estimates (~12 months), with a high consistency rate of 87.5%, suggesting that GPCC and GPCP are both reliable for global hydrological applications. The largest discrepancy in mean DRT estimation (0.1 months) was observed in the polar (E) zone, with no significant difference in the snow (D) zone. The consistency of the two precipitation products differed by less than 1% across all the climate zones. These results for the precipitation products highlight the robustness of these products across the diverse climate zones.*

*For the TWS products, the global mean DRT estimation using JPL mascon (13.8 months) was 2.6 months higher than that of G3P (11.4 months). G3P exhibited 5.0% higher global consistency (90.0%) than did JPL mascon (85.0%), suggesting that it is better suited for analysing hydrological drought characteristics, particularly in regions with extreme climate conditions, such as the polar (E) and arid (B) zones, where G3P outperformed JPL mascon by 7.0% in consistency. The significant disparity in mean DRT estimation in the polar zone (13.2 months for G3P versus 18.9 months for JPL mascon) highlights the challenges of accurately representing water storage dynamics in high-latitude regions, possibly due to differences in how the two products handle ice and snow storage variability. Conversely, the smallest differences in DRT estimates and consistency in the equatorial (A) zone suggest that both TWS*

*products perform effectively in regions with stable precipitation patterns. These findings reveal that while both precipitation products perform similarly, G3P outperforms JPL mascon in consistency and alignment with TWS and precipitation based DRT. This suggests that G3P may provide a more accurate representation of terrestrial water storage dynamics in diverse climate zones.*

*Potential limitations of this study include the reliance on specific TWS and precipitation products, which may not fully capture uncertainties associated with data retrieval and processing. The study also does not account for external factors such as anthropogenic water use or land cover/use changes, which can influence DRT estimates. Future studies could benefit from integrating additional datasets and models to provide a more comprehensive understanding of these dynamics.”*

**Conclusion:**

The manuscript appears to provide an innovative and interesting method for evaluating precipitation products but neglects the main conclusions and their implications regarding the evaluation of GPCC, GPCP, G3P, and JPL Mascon. The conclusion should explicitly state which products show the best performance and in which contexts, along with the broader implications of these findings.

Authors: In this revised version, we have strengthened the conclusion to explicitly state which products demonstrate the best performance and under which contexts. Additionally, we have highlighted the broader implications of these findings to enhance the overall clarity and impact of the study. We have revised the conclusion section as follows:

*“TWS changes from one time epoch to another as observed by satellite gravimetry are closely related to the precipitation amount occurring during that time interval. The novel observing concept realized by the GRACE and GRACE-FO missions thus provides a unique opportunity to evaluate the frequently used global precipitation products on monthly or longer timescales. GRACE/GRACE-FO directly provides water storage anomalies, offering a novel approach to characterize drought by assessing the storage deficit. The time required for drought recovery can be directly derived from the temporal evolution of these deficits (Singh et al., 2021), enabling the measurement of both drought duration and severity.*

*Our assessments reveal that both GPCC and GPCP products exhibited not only similar DRT estimates but also comparable consistency globally and across all the Köppen-Geiger climate zones. However, as noted, GPCP's reliance on satellite data enhances its utility in data-sparse regions, making it a more versatile choice in such contexts. However, this advantage does not extend to regions with dense in situ networks, where the inclusion of satellite data does not improve its performance.*

*For TWS products, the mean DRT estimates from JPL mascon were, on average, 2.6 months higher than those from G3P, globally and across all the Köppen-Geiger climate zones. However, G3P showed slightly higher consistency in the DRT estimates (5.0% difference, globally) than did JPL mascon. Furthermore, G3P demonstrated greater consistency than JPL mascon across all the Köppen-Geiger climate zones. These findings highlight G3P's reliability for applications requiring precise water storage anomaly data, such as drought monitoring.*

*The results of our study underline the potential value of GRACE/GRACE-FO for hydrometeorological research due to the strong relationship between precipitation and TWS changes. Its global coverage (albeit its rather low spatial resolution) allows the testing of different precipitation products not only those derived from combinations of satellite and in situ observations (as done in this study) but also from numerical weather prediction models and global atmospheric reanalyses. Both NASA and the European Space Agency (ESA) are currently working on future satellite gravity missions with even more precise sensors and different orbital configurations to further enhance the quality of satellite gravimetry products for hydrological applications."*

**3. Minor comments**

**Line 24:** The term "performance" is too vague. Please specify what aspect of performance is being referred to.

Author: To clarify the "performance" term, we have now revised the following text in the manuscript (Abstract):

*"This study also evaluates the drought monitoring performance of G3P and JPL mascon Total Water Storage (TWS) monthly-solutions from the Gravity Recovery and Climate Experiment (GRACE) and GRACE Follow-On (GRACE-FO) satellite missions."*

**Line 27:** The phrase "difference between the two methods" is too vague. Provide a concise explanation of what the difference entails.

Authors: To clarify the "difference between the two methods", we have now revised the following text in the manuscript (Abstract):

*"The current study employed two methods to estimate DRT and evaluated the consistency of DRT estimations by calculating the time difference in DRT values derived from the two methods."*

**Figure 5:** Avoid showing panels (b) and (d) if there are no precipitation data used in the DRT estimation and they have no impact on the results.

Authors: To clarify the Figure 5, we have now revised the following text in the Section "*3.2 DRT Estimates*":

*"The precipitation data are not utilized in calculating DRT estimates based on the storage deficit method (Fig. 5). However, they are used in the masking procedure for regions with weak or no linear relationship between cdPA and dTWSA. Although the unmasked regions have identical DRT values, the differences between Fig. 5a and 5b, as well as Fig. 5c and 5d, arise from whether a region is masked or unmasked. Consequently, these figures are not identical."*

**Figure 8:** If the DRT values presented in this figure are derived as an average between DRT calculated from storage deficit and DRT calculated from required precipitation, this should be clarified explicitly in the caption.

Authors: To clarify the explanation of the figure, we have now revised the following text in the manuscript (Figure 8):

*"Representation of the consistency in DRT estimates (the class of the time difference in DRT values between two methods, see Table 1), obtained from the different dTWSA (i.e., G3P, JPL mascon) and cdPA (i.e., GPCP, GPCC) datasets. (a) consistency using dTWSA from G3P and cdPA from GPCP (G3P–GPCP), and differences in consistency class relative to G3P–GPCP for (b) G3P–GPCC, (c) JPL mascon–GPCP, and (d) JPL mascon–GPCC."*

---

## Author Response (AR3)

I thank the authors for their thoughtful revisions and clarifications. The objectives, results, and conclusions are now more clearly presented, and the addition of a discussion section helps better contextualize the findings. The manuscript is substantially improved.

However, before recommending acceptance, I suggest a minor revision to improve the discussion of the method's limitations.

The current discussion of limitations remains somewhat superficial. For example, the authors mention line 496 that external factors, such as anthropogenic water use or land cover changes, could influence DRT estimates, and that uncertainties in data retrieval and processing exist. While these points are valid, they are presented quite briefly. A more reflective discussion of these limitations, including potential impacts on the linearity assumption and the overall reliability of DRT estimation, would provide a deeper understanding of the method's scope and constraints.

In particular, it would be helpful to briefly address:

• The role of unobserved hydrological fluxes, such as evapotranspiration and runoff, and how they might affect the relationship between precipitation and storage variations.

• The spatial scale mismatch: while precipitation data are aggregated to the GRACE resolution, a brief discussion of how differences in signal characteristics may still affect comparability would be useful.

• Regression behavior: The authors present a global map of $\beta_1$ values, which is informative, but further clarification is needed on how to interpret high values (e.g., >2 or >3). High $\beta_1$ values could indicate non-linearity in the precipitation-storage relationship or potential methodological issues, such as data noise or model misfit. A brief discussion on this would help assess the model's reliability and its applicability to different regions.

I am not requesting any new analysis, only a more explicit discussion of these points in the text.

Authors: We thank the reviewer for the constructive comments. In this revised version, we have substantially strengthened the limitations of the method section. We have revised the section as follows:

"While this study demonstrates the utility of DRT estimates derived from precipitation and GRACE/GRACE-FO TWSA data for evaluating global datasets, it is essential to discuss certain methodological and data-related limitations to appropriately contextualize the findings. First, we assumed a linear relationship between cdPA and dTWSA. However, this dynamic relationship may be disrupted by anthropogenic activities (e.g., groundwater extraction, dam construction, deforestation, and urbanization) as well as natural processes (e.g., evapotranspiration and runoff) which can modify the hydrological response independently of precipitation dynamics. These factors may delay the transfer of precipitation into storage components or reduce the volume ultimately contributing to storage. Second, uncertainties inherent to GRACE/GRACE-FO data processing and precipitation products, stemming from sensor characteristics or model parameterizations, may introduce noise or systematic biases

*into DRT estimates and their consistency. Third, the simplified water balance framework assumes stable partitioning of precipitation into evapotranspiration and runoff. However, temporal variability in these processes, driven by factors such as temperature, vegetation dynamics, or soil moisture conditions, may weaken the cdPA and dTWSA relationship, particularly in energy-limited regions (e.g., high-evapotranspiration zones) or areas highly sensitive to runoff (e.g., snowmelt-dominated basins). Fourth, a spatial scale mismatch remains between datasets: GRACE/GRACE-FO's coarse spatial resolution smooths fine-scale TWS variability, while spatially aggregated precipitation data may obscure localized hydrometeorological events (e.g., intense convective rainfall), thereby affecting the precipitation–TWS relationship in regions characterized by complex topography or localized weather systems. Finally, the occurrence of high regression coefficients ($\beta_1 > 2$) highlights the sensitivity of storage changes to precipitation inputs and suggests the presence of unmodeled nonlinearities or time lags in the hydrological response. The high $\beta_1$ values, indicating rapid precipitation removal via runoff or evapotranspiration, may reflect limitations in the linear model's ability to capture delayed storage responses or nonlinearities in specific hydrological regimes. Despite these limitations, this study provides a valuable framework for assessing global precipitation and TWS products via the hydrological drought characteristics."*